# Identifiability Analysis of Linear ODE Systems with Hidden Confounders

**Yuanyuan Wang**
The University of Melbourne
yuanyuanw2@student.unimelb.edu.au

**Biwei Huang**
University of California, San Diego
bih007@ucsd.edu

**Wei Huang**
The University of Melbourne
wei.huang@unimelb.edu.au

**Xi Geng**
The University of Melbourne
xi.geng@unimelb.edu.au

**Mingming Gong** *
The University of Melbourne
mingming.gong@unimelb.edu.au

## Abstract

The identifiability analysis of linear Ordinary Differential Equation (ODE) systems is a necessary prerequisite for making reliable causal inferences about these systems. While identifiability has been well studied in scenarios where the system is fully observable, the conditions for identifiability remain unexplored when latent variables interact with the system. This paper aims to address this gap by presenting a systematic analysis of identifiability in linear ODE systems incorporating hidden confounders. Specifically, we investigate two cases of such systems. In the first case, latent confounders exhibit no causal relationships, yet their evolution adheres to specific functional forms, such as polynomial functions of time $t$. Subsequently, we extend this analysis to encompass scenarios where hidden confounders exhibit causal dependencies, with the causal structure of latent variables described by a Directed Acyclic Graph (DAG). The second case represents a more intricate variation of the first case, prompting a more comprehensive identifiability analysis. Accordingly, we conduct detailed identifiability analyses of the second system under various observation conditions, including both continuous and discrete observations from single or multiple trajectories. To validate our theoretical results, we perform a series of simulations, which support and substantiate our findings.

## 1 Introduction

Understanding the dynamics of systems governed by Ordinary Differential Equations (ODEs) is fundamental in various scientific disciplines, from physics [9, 23, 24, 47], biology [16, 27, 29, 33, 36] to economics [13, 38, 39, 43]. These ODE systems provide a natural framework for modeling causal relationships among system variables, enabling us to make reliable interpretations and interventions [25, 30, 31]. Central to unraveling the causal mechanisms of such systems is the concept of identifiability analysis, which aims to uncover conditions under which system parameters can be uniquely determined from error-free observations. Identifiability is crucial for ensuring reliable parameter estimates, thereby guaranteeing reliable causal inferences about the system [41]. The motivation for our research on the identifiability analysis of ODE systems arises from the necessity of making reliable causal inferences about these systems.

---

*Corresponding author.

38th Conference on Neural Information Processing Systems (NeurIPS 2024).

Our research focuses on the homogeneous linear ODE system, represented as:
$$\dot{\boldsymbol{x}}(t) = A\boldsymbol{x}(t), \quad \boldsymbol{x}(0) = \boldsymbol{x}_0, \tag{1}$$
where $t \in [0, \infty)$ denotes time, $\boldsymbol{x}(t) \in \mathbb{R}^d$ represents the system's state at time $t$, $\dot{\boldsymbol{x}}(t)$ denotes the first derivative of $\boldsymbol{x}(t)$ w.r.t. time, and $\boldsymbol{x}_0$ represents the initial condition of the system. The solution (trajectory) of the system, denoted as $\boldsymbol{x}(t; \boldsymbol{x}_0, A)$ for $t \in [0, \infty)$, is a single $d$-dimensional trajectory initialized with $\boldsymbol{x}_0$.

Existing literature has extensively examined the identifiability of linear ODE systems under the assumption of complete observability, where all state variables are directly observable [5, 14, 15, 17, 28, 34, 42]. Specifically, researchers have investigated identifiability of the ODE system (1) from a single whole trajectory [28, 34], and extended analysis to discrete observations sampled from the trajectory [42]. However, practical scenarios often entail systems with latent variables, rendering them not entirely observable. In this paper, we explore the identifiability analysis of this ODE system under latent confounders, particularly examining cases where no causal relationships exist from observable variables to latent variables, a commonly assumed condition in causality analysis with hidden variables [10, 11, 20, 22, 44, 45].

In this paper, we focus on two scenarios:

1. **Independent latent confounders:** Latent variables exhibit no causal relationships among themselves, leading to the following linear ODE system:
$$\begin{bmatrix} \dot{\boldsymbol{x}}(t) \\ \dot{\boldsymbol{z}}(t) \end{bmatrix} = \begin{bmatrix} A & B \\ \mathbf{0} & \mathbf{0} \end{bmatrix} \begin{bmatrix} \boldsymbol{x}(t) \\ \boldsymbol{z}(t) \end{bmatrix} + \begin{bmatrix} \mathbf{0} \\ \boldsymbol{f}(t) \end{bmatrix}, \quad \begin{bmatrix} \boldsymbol{x}(0) \\ \boldsymbol{z}(0) \end{bmatrix} = \begin{bmatrix} \boldsymbol{x}_0 \\ \boldsymbol{z}_0 \end{bmatrix}. \tag{2}$$

2. **Causally related latent confounders:** Latent variables exhibit causal relationships among themselves, specifically, they follow a DAG structure, represented as:
$$\begin{bmatrix} \dot{\boldsymbol{x}}(t) \\ \dot{\boldsymbol{z}}(t) \end{bmatrix} = \begin{bmatrix} A & B \\ \mathbf{0} & G \end{bmatrix} \begin{bmatrix} \boldsymbol{x}(t) \\ \boldsymbol{z}(t) \end{bmatrix}, \quad \begin{bmatrix} \boldsymbol{x}(0) \\ \boldsymbol{z}(0) \end{bmatrix} = \begin{bmatrix} \boldsymbol{x}_0 \\ \boldsymbol{z}_0 \end{bmatrix}. \tag{3}$$

In these two ODE systems, $\boldsymbol{x}(t) \in \mathbb{R}^d$ denotes the state of observable variables $\boldsymbol{x} = (x_1, x_2, \ldots, x_d)$, while $\boldsymbol{z}(t) \in \mathbb{R}^p$ denotes the state of latent variables $\boldsymbol{z} = (z_1, z_2, \ldots, z_p)$. Example causal structures of these two ODE systems are illustrated in Figure 1. It is noteworthy that the structure may include cycles and self-loops within the observable variables. Additionally, two real-world examples are provided in Appendix B.

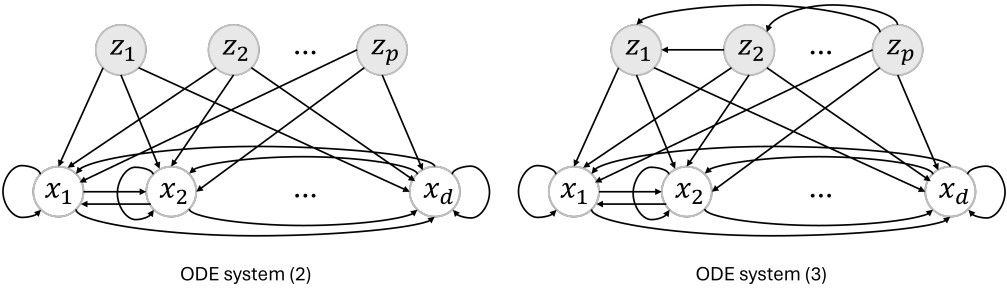

ODE system (2)                    ODE system (3)

Figure 1: Example causal structures of the ODE system (2) and (3).

This paper provides an identifiability analysis for the ODE system (2) under specific latent variable evolutions, such as polynomial functions of time $t$. Additionally, we conduct a systematic identifiability analysis of the ODE system (3) when the causal structure of the latent variables can be described by a DAG.

## 2 Background

### 2.1 Causal interpretation of the ODE system

When an ODE system describes the underlying causal mechanisms governing a dynamic system, it provides a natural framework for modeling causal relationships among system variables. The causal

structure inherent in such systems can be directly read off [25, 31]. For instance, in the ODE system (1), where the $ij$-th entry of the parameter matrix $A$ is denoted as $A_{ij}$, the presence of $A_{ij} \neq 0$ signifies that the derivative of $x_i(t)$ is influenced by $x_j(t)$, thus indicating a causal link from $x_j$ to $x_i$. Here, $x_i$ denotes the $i$-th variable of the ODE system (1), and $x_i(t)$ represents its state at time $t$. Since the right hand side of the ODE system (1) does not explicitly depend on time $t$, the causal structure of this ODE system is time-invariant.

An essential prerequisite for reliably inferring the causal structure and effects of an ODE system, for purposes of interpretation or intervention, is the identifiability analysis of such systems. To underscore this necessity, we provide an illustrative example. Consider the ODE system (3). Set

$$\boldsymbol{x}_0 = \begin{bmatrix} 1 \\ 1 \end{bmatrix}, \quad \boldsymbol{z}_0 = \begin{bmatrix} 1 \\ 1 \end{bmatrix}, \quad B = \begin{bmatrix} 1 & 1 \\ 1 & 1 \end{bmatrix}, \quad G = \begin{bmatrix} 0 & 1 \\ 0 & 0 \end{bmatrix},$$

$$A = \begin{bmatrix} 1 & 0 \\ 0 & 1 \end{bmatrix}, \quad A' = \begin{bmatrix} 0 & 1 \\ 1 & 0 \end{bmatrix}, \quad M = \begin{bmatrix} A & B \\ \mathbf{0} & G \end{bmatrix}, \quad M' = \begin{bmatrix} A' & B \\ \mathbf{0} & G \end{bmatrix}.$$

Calculations reveal that the solutions (trajectory) of the ODE system (3) with parameter matrices $M$ or $M'$ are identical, i.e.,

$$\begin{bmatrix} \boldsymbol{x}(t) \\ \boldsymbol{z}(t) \end{bmatrix} = e^{Mt} \begin{bmatrix} \boldsymbol{x}_0 \\ \boldsymbol{z}_0 \end{bmatrix} = e^{M't} \begin{bmatrix} \boldsymbol{x}_0 \\ \boldsymbol{z}_0 \end{bmatrix}.$$

This indicates that using observations sampled from this trajectory to estimate parameter matrix $M$ may end up yielding $M'$, which exhibits a fundamentally distinct causal relationship between $x_1$ and $x_2$, see Figure 2. This discrepancy in parameter estimation, wherein $M'$ is obtained instead

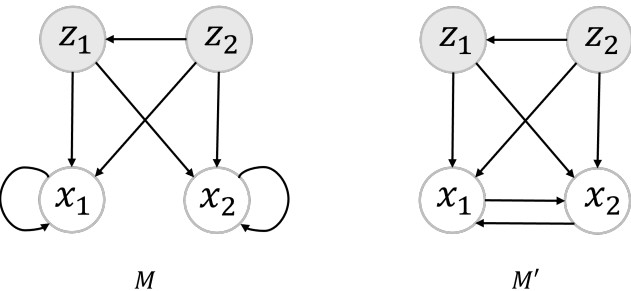

Figure 2: Causal structures of the ODE system (3) with parameter matrix $M$ and $M'$.

of the true underlying parameter matrix $M$, may lead to misleading interpretations and causal inferences, potentially influencing decision-making, particularly regarding interventions. For instance, intervention with $x_1(t) = 1$, under the true underlying parameter matrix $M$, yields the trajectory $x_2(t) = 4e^t - t - 3$ (post-intervention), whereas under matrix $M'$, the trajectory becomes $x_2(t) = t^2/2 + 3t + 1$ (post-intervention). Detailed calculations are provided in Appendix C.

## 2.2 Identifiability analysis of the linear ODE system (1)

The identifiability analysis of the ODE system (1) has been well studied. Here, we present a fundamental definition and theorem essential for understanding identifiability in the ODE system (1). Denoting its solution as $\boldsymbol{x}(t; \boldsymbol{x}_0, A)$, it is noteworthy that the system is fully observable, without latent variables interacting with it. We present the identifiability definition and theorem as follows.

**Definition 2.1.** *For $\boldsymbol{x}_0 \in \mathbb{R}^d$, $A \in \mathbb{R}^{d \times d}$, the ODE system (1) is said to be $(\boldsymbol{x}_0, A)$-identifiable, if for all $\boldsymbol{x}_0' \in \mathbb{R}^d$ and all $A' \in \mathbb{R}^{d \times d}$, with $(\boldsymbol{x}_0, A) \neq (\boldsymbol{x}_0', A')$, it holds that $\boldsymbol{x}(\cdot; \boldsymbol{x}_0, A) \neq \boldsymbol{x}(\cdot; \boldsymbol{x}_0', A')$.[2]*

**Lemma 2.1.** *For $\boldsymbol{x}_0 \in \mathbb{R}^d$, $A \in \mathbb{R}^{d \times d}$, the ODE system (1) is $(\boldsymbol{x}_0, A)$-identifiable if and only if condition **A0** is satisfied.*

    ***A0*** *the set of vectors $\{\boldsymbol{x}_0, A\boldsymbol{x}_0, \ldots, A^{d-1}\boldsymbol{x}_0\}$ is linearly independent.*

---

[2]$\boldsymbol{x}(\cdot; \boldsymbol{x}_0, A) = \{\boldsymbol{x}(t; \boldsymbol{x}_0, A) : 0 \leqslant t < \infty\}$, this inequation means that there exists at least one $t \geqslant 0$ such that $\boldsymbol{x}(t; \boldsymbol{x}_0, A) \neq \boldsymbol{x}(t; \boldsymbol{x}_0', A')$.

Definition 2.1 and Theorem 2.1 are adapted from [42, Definition 1] and [42, Lemma2]. We use $\boldsymbol{x}_0'$ and $A'$ to distinguish other system parameters from the true system parameters $\boldsymbol{x}_0$ and $A$; $\boldsymbol{x}_0'$ and $A'$ can represent any $d$-dimensional initial conditions and any $d \times d$ parameter matrices, respectively. Here instead of describing a collective property of a set of systems, we describe an intrinsic property of a single system with parameters $(\boldsymbol{x}_0, A)$. In practice, the aim is to ascertain whether the true underlying system parameter $(\boldsymbol{x}_0, A)$ is uniquely determined by observations. Hence, $(\boldsymbol{x}_0, A)$-identifiability offers a more intuitive description of the identifiability of the ODE system from a practical perspective.

From a geometric perspective, condition **A0** stated in Lemma 2.1 indicates that the initial condition $\boldsymbol{x}_0$ is not contained in an $A$-invariant **proper** subspace of $\mathbb{R}^d$. Intuitively, this means the trajectory of this system started from $\boldsymbol{x}_0$ spans the entire $d$-dimensional state space. That is, our observations cover information on all dimensions of the state space, thus rendering the identifiability of the system. Additionally, condition **A0** is generic, as noted in [41], meaning that the set of system parameters violating this condition has Lebesgue measure zero. Thus, condition **A0** is satisfied for almost all combinations of $\boldsymbol{x}_0$ and $A$.

# 3 Identifiability analysis of the linear ODE system (2)

In this section, we present the identifiability condition for the linear ODE system (2). We consider the function $\boldsymbol{f}(t)$ in (2) as a specific function of time $t$. Here we first define $\boldsymbol{f}(t)$ as a $r$-degree polynomial function of time $t$, expressed as follows:

$$\boldsymbol{f}(t) = \sum_{k=0}^{r} \boldsymbol{v}_k t^k, \quad \boldsymbol{v}_k \in \mathbb{R}^p. \tag{4}$$

Simple calculations show that

$$\boldsymbol{z}(t) = \sum_{k=0}^{r} \frac{\boldsymbol{v}_k}{k+1} t^{k+1} + \boldsymbol{z}_0.$$

Thus,

$$\dot{\boldsymbol{x}}(t) = A\boldsymbol{x}(t) + B\boldsymbol{z}(t) = A\boldsymbol{x}(t) + \sum_{k=0}^{r} \frac{B\boldsymbol{v}_k}{k+1} t^{k+1} + B\boldsymbol{z}_0. \tag{5}$$

We denote the unknown parameters of the ODE system (2) as $\boldsymbol{\theta}$, specifically, $\boldsymbol{\theta} := (\boldsymbol{x}_0, \boldsymbol{z}_0, A, B, \{\boldsymbol{v}_k\}_0^r)$, where $\{\boldsymbol{v}_k\}_0^r$ denotes all the $\boldsymbol{v}_k$'s for $k = 0, \ldots, r$. Let $[\boldsymbol{x}^T(t; \boldsymbol{\theta}), \boldsymbol{z}^T(t; \boldsymbol{\theta})]^T$ denote the solution of the ODE system (2). It is important to note that under our hidden variables setting, only $\boldsymbol{x}(t; \boldsymbol{\theta})$ is observable. Based on Equation (5), we present the following identifiability definition.

**Definition 3.1.** *For $\boldsymbol{x}_0 \in \mathbb{R}^d, \boldsymbol{z}_0 \in \mathbb{R}^p, A \in \mathbb{R}^{d \times d}, B \in \mathbb{R}^{d \times p}$ and $\{\boldsymbol{v}_k\}_0^r \in \mathbb{R}^p$, for all $\boldsymbol{x}_0' \in \mathbb{R}^d$, all $\boldsymbol{z}_0' \in \mathbb{R}^p$, all $A' \in \mathbb{R}^{d \times d}$, all $B' \in \mathbb{R}^{d \times p}$, and all $\{\boldsymbol{v}_k'\}_0^r \in \mathbb{R}^p$, we denote $\boldsymbol{\theta}' := (\boldsymbol{x}_0', \boldsymbol{z}_0', A', B', \{\boldsymbol{v}_k'\}_1^r)$, we say the ODE system (2) is $\boldsymbol{\theta}$-identifiable: if $(\boldsymbol{x}_0, A, B\boldsymbol{z}_0, \{B\boldsymbol{v}_k\}_0^r) \neq (\boldsymbol{x}_0', A', B'\boldsymbol{z}_0', \{B'\boldsymbol{v}_k'\}_0^r)$, it holds that $\boldsymbol{x}(\cdot; \boldsymbol{\theta}) \neq \boldsymbol{x}(\cdot; \boldsymbol{\theta}')$.*

In the ODE system (2), where only variables $\boldsymbol{x}$ are observable, we will, with some terminological leniency, refer to $\boldsymbol{x}(\cdot; \boldsymbol{\theta})$ as the trajectory of the ODE system (2) with parameters $\boldsymbol{\theta}$. According to Definition 3.1, if the ODE system (2) with a polynomial $\boldsymbol{f}(t)$ is $\boldsymbol{\theta}$-identifiable, then the trajectory of the system can uniquely determine the values of $(\boldsymbol{x}_0, A, B\boldsymbol{z}_0, \{B\boldsymbol{v}_k\}_0^r)$. This determination is sufficient to identify the causal relationships between observable variables $\boldsymbol{x}$ as described by Equation (5). Consequently, one can safely intervene in the observable variables of the ODE system and make reliable causal inferences, despite the fact that matrix $B$ cannot be identified under this definition.

**Theorem 3.1.** *For $\boldsymbol{x}_0 \in \mathbb{R}^d, \boldsymbol{z}_0 \in \mathbb{R}^p, A \in \mathbb{R}^{d \times d}, B \in \mathbb{R}^{d \times p}, \{\boldsymbol{v}_k\}_0^r \in \mathbb{R}^p$, the ODE system (2) is $\boldsymbol{\theta}$-identifiable if and only if assumption A1 is satisfied.*

**A1** *the set of vectors $\{\boldsymbol{\beta}, A\boldsymbol{\beta}, \ldots, A^{d-1}\boldsymbol{\beta}\}$ is linearly independent, where $\boldsymbol{\beta} = A^{r+1}(A\boldsymbol{x}_0 + B\boldsymbol{z}_0) + \sum_{j=0}^{r} j! A^{r-j} B\boldsymbol{v}_j$, and $j!$ denotes the factorial of $j$.*

The proof of Theorem 3.1 can be found in Appendix D.1. Condition **A1** is both sufficient and necessary, indicating, from a geometric perspective, that the vector $\boldsymbol{\beta}$ is not contained in an $A$-invariant proper subspace of $\mathbb{R}^d$ [34, Lemma 3.1].

The key point of the proof is the introduction of an augmented state $\boldsymbol{y}(t) = [\boldsymbol{x}^T(t), 1, t, t^2, \ldots, t^{r+1}]^T$ with a corresponding ODE system:

$$\dot{\boldsymbol{y}}(t) = \underbrace{\begin{bmatrix} A & B\boldsymbol{z}_0 & B\boldsymbol{v}_0 & \ldots & B\boldsymbol{v}_{r-1}/r & B\boldsymbol{v}_r/(r+1) \\ \mathbf{0}_d & 0 & 0 & \ldots & 0 & 0 \\ \mathbf{0}_d & 1 & 0 & \ldots & 0 & 0 \\ \vdots & \vdots & \vdots & \ddots & \vdots & \vdots \\ \mathbf{0}_d & 0 & 0 & \ldots & r+1 & 0 \end{bmatrix}}_{\text{denoted as } F} \boldsymbol{y}(t), \tag{6}$$

$$\boldsymbol{y}(0) = [\boldsymbol{x}_0^T, 1, 0, \ldots, 0]^T := \boldsymbol{y}_0,$$

where $\mathbf{0}_d$ is a $d$-dimensional zero row vector, and matrix $F \in \mathbb{R}^{(d+r+2) \times (d+r+2)}$. The ODE system (6) is a homogeneous linear ODE system analogous to (1) but with fully observable variables $\boldsymbol{y}$. In other words, we transform our system of interest, (2), which includes hidden confounders, into a fully observable ODE system (6). This allows us to leverage existing identifiability results for homogeneous linear ODE systems, specifically Lemma 2.1, to derive the identifiability condition for the ODE system (2).

Based on this approach, if the state of the hidden variables $\boldsymbol{z}(t)$, as determined by the function $\boldsymbol{f}(t)$ in the ODE system (2), can be described by some linear combinations of observable functions of time $t$, then the identifiability condition of the ODE system (2) can be derived. For an illustration, in the Appendix E, we provide identifiability conditions for the ODE system (2) when $\boldsymbol{f}(t) = \boldsymbol{v}e^t$ and $\boldsymbol{f}(t) = \boldsymbol{v}_1 sin(t) + \boldsymbol{v}_2 cos(t)$. While we do not enumerate all functions $\boldsymbol{f}(t)$ that meet this condition, our primary objective is to demonstrate a method for deriving the identifiability condition for the ODE (2) when the evolution of its hidden variables conforms to certain specific functions. Researchers can apply this approach to find appropriate functions $\boldsymbol{f}(t)$ according to their specific requirements.

## 4 Identifiability analysis of the linear ODE system (3)

In this section, we extend the identifiability analysis to linear ODE systems with causally related latent confounders. Specifically, we assume that the causal structure of latent variables satisfies the following latent DAG assumption.

> **Latent DAG**: the causal structure of latent variables can be described by a DAG.

The DAG assumption is commonly employed in causality studies [10, 11, 22, 26, 40, 44]. Under the latent DAG assumption, the matrix $G$ can be permuted to be a strictly upper triangular matrix, i.e., an upper triangular matrix with zeros along the main diagonal [11, 19]. Without loss of generality, we set $G$ as a strictly upper triangular matrix.

Since $G$ is a strictly upper triangular matrix, by the Cayley–Hamilton theorem [35], $G$ is a nilpotent matrix with an index $\leqslant p$. Consequently, $G^k = 0$ for all $k \geqslant p$.

Based on [34, 37], the solution of $\boldsymbol{z}(t)$ can be expressed as:

$$\boldsymbol{z}(t) = e^{Gt}\boldsymbol{z}_0 = \sum_{k=0}^{\infty} \frac{G^k \boldsymbol{z}_0}{k!} t^k = \sum_{k=0}^{p-1} \frac{G^k \boldsymbol{z}_0}{k!} t^k.$$

Thus,

$$\dot{\boldsymbol{x}}(t) = A\boldsymbol{x}(t) + B\boldsymbol{z}(t) = A\boldsymbol{x}(t) + \sum_{k=0}^{p-1} \frac{BG^k \boldsymbol{z}_0}{k!} t^k. \tag{7}$$

We observe that Equation (7) has the same function form as Equation (5), but with different coefficients (system parameters) for the polynomial of time $t$. Therefore, the ODE system (3) under the latent DAG assumption can be considered a more complex version of the ODE system (2) when $\boldsymbol{f}(t)$ follows a polynomial function of time $t$. Since the ODE system (3) incorporates causally related latent confounders, which is a more interesting and practical case, we will provide a more comprehensive identifiability analysis of the ODE system (3). The derived identifiability results can be easily generated to the case of the ODE system (2).

## 4.1 Identifiability condition from a single whole trajectory

We denote the unknown parameters of the ODE system (3) as $\boldsymbol{\eta}$, that is, $\boldsymbol{\eta} := (\boldsymbol{x}_0, \boldsymbol{z}_0, A, B, G)$. We further denote the solution of the ODE system (3) as $[\boldsymbol{x}^T(t; \boldsymbol{\eta}), \boldsymbol{z}^T(t; \boldsymbol{\eta})]^T$; note that under our latent variables setting, only $\boldsymbol{x}(t; \boldsymbol{\eta})$ is observable. Thus, based on Equation (7), we present the following identifiability definition.

**Definition 4.1.** *For $\boldsymbol{x}_0 \in \mathbb{R}^d, \boldsymbol{z}_0 \in \mathbb{R}^p, A \in \mathbb{R}^{d \times d}, B \in \mathbb{R}^{d \times p}$ and $G \in \mathbb{R}^{p \times p}$, under the latent DAG assumption, for all $\boldsymbol{x}_0' \in \mathbb{R}^d$, all $\boldsymbol{z}_0' \in \mathbb{R}^p$, all $A' \in \mathbb{R}^{d \times d}$, all $B' \in \mathbb{R}^{d \times p}$, and all $G' \in \mathbb{R}^{p \times p}$, we denote $\boldsymbol{\eta}' := (\boldsymbol{x}_0', \boldsymbol{z}_0', A', B', G')$, we say the ODE system (3) is $\boldsymbol{\eta}$-identifiable: if $(\boldsymbol{x}_0, A, B\boldsymbol{z}_0, BG\boldsymbol{z}_0, \ldots, BG^{p-1}\boldsymbol{z}_0) \neq (\boldsymbol{x}_0', A', B'\boldsymbol{z}_0', B'G'\boldsymbol{z}_0', \ldots, B'G'^{p-1}\boldsymbol{z}_0')$, it holds that $\boldsymbol{x}(\cdot; \boldsymbol{\eta}) \neq \boldsymbol{x}(\cdot; \boldsymbol{\eta}')$.*

Similar to the case of the ODE system (2), we refer to $\boldsymbol{x}(\cdot; \boldsymbol{\eta})$ as the trajectory of the ODE system (3) with parameters $\boldsymbol{\eta}$. Definition 4.1 defines the identifiability of the ODE system (3) from a single whole trajectory $\boldsymbol{x}(\cdot; \boldsymbol{\eta})$. Once the ODE system (3) is $\boldsymbol{\eta}$-identifiable, the causal relationships among the observable variables $\boldsymbol{x}$ can be determined through Equation (7). We then establish the condition for the identifiability of the ODE system (3) based on Definition 4.1.

**Theorem 4.1.** *For $\boldsymbol{x}_0 \in \mathbb{R}^d, \boldsymbol{z}_0 \in \mathbb{R}^p, A \in \mathbb{R}^{d \times d}, B \in \mathbb{R}^{d \times p}$ and $G \in \mathbb{R}^{p \times p}$, under the latent DAG assumption, the ODE system (3) is $\boldsymbol{\eta}$-identifiable if and only if assumption **B1** is satisfied.*

> **B1:** *the set of vectors $\{\boldsymbol{\gamma}, A\boldsymbol{\gamma}, \ldots, A^{d-1}\boldsymbol{\gamma}\}$ is linearly independent, where $\boldsymbol{\gamma} = A^p\boldsymbol{x}_0 + \sum_{j=0}^{p-1} A^{p-1-j}BG^j\boldsymbol{z}_0$.*

The proof of Theorem 4.1 can be found in Appendix D.2. Condition **B1** is both sufficient and necessary, and from a geometric perspective, it indicates that the vector $\boldsymbol{\gamma}$ is not contained in an $A$-invariant proper subspace of $\mathbb{R}^d$ [34, Lemma 3.1].

## 4.2 Identifiability condition from discrete observations sampled from a single trajectory

In practice, we often have access only to a sequence of discrete observations sampled from a trajectory rather than knowing the whole trajectory. Therefore, we also derive the identifiability conditions under the scenario where only discrete observations from a trajectory are available. Firstly, we extend the identifiability definition of the ODE system (3) as follows.

**Definition 4.2.** *For $\boldsymbol{x}_0 \in \mathbb{R}^d, \boldsymbol{z}_0 \in \mathbb{R}^p, A \in \mathbb{R}^{d \times d}, B \in \mathbb{R}^{d \times p}$ and $G \in \mathbb{R}^{p \times p}$. For any $n \geqslant 1$, let $t_j, j = 1, \ldots, n$ be any $n$ time points and $\boldsymbol{x}_j := \boldsymbol{x}(t_j; \boldsymbol{\eta})$ be the error-free observation of the trajectory $\boldsymbol{x}(\cdot; \boldsymbol{\eta})$ at time $t_j$. Under the latent DAG assumption, we say the ODE system (3) is $\boldsymbol{\eta}$-identifiable from $\boldsymbol{x}_1, \ldots, \boldsymbol{x}_n$, if for all $\boldsymbol{x}_0' \in \mathbb{R}^d$, all $\boldsymbol{z}_0' \in \mathbb{R}^p$, all $A' \in \mathbb{R}^{d \times d}$, all $B' \in \mathbb{R}^{d \times p}$, and all $G' \in \mathbb{R}^{p \times p}$ with $(\boldsymbol{x}_0, A, B\boldsymbol{z}_0, BG\boldsymbol{z}_0, \ldots, BG^{p-1}\boldsymbol{z}_0) \neq (\boldsymbol{x}_0', A', B'\boldsymbol{z}_0', B'G'\boldsymbol{z}_0', \ldots, B'G'^{p-1}\boldsymbol{z}_0')$, it holds that $\exists j \in \{1, \ldots, n\}$ such that $\boldsymbol{x}(t_j; \boldsymbol{\eta}) \neq \boldsymbol{x}(t_j; \boldsymbol{\eta}')$.*

Definition 4.2 defines the identifiability of the ODE system (3) from $n$ observations sampled from the trajectory $\boldsymbol{x}(\cdot; \boldsymbol{\eta})$. Then we establish the condition for the identifiability of the ODE system (3) from discrete observations based on Definition 4.2.

**Theorem 4.2.** *For $\boldsymbol{x}_0 \in \mathbb{R}^d, \boldsymbol{z}_0 \in \mathbb{R}^p, A \in \mathbb{R}^{d \times d}, B \in \mathbb{R}^{d \times p}$ and $G \in \mathbb{R}^{p \times p}$. We define new observation $\boldsymbol{y}_j := [\boldsymbol{x}_j^T, 1, t_j, t_j^2, \ldots, t_j^{p-1}]^T \in \mathbb{R}^{d+p}$, for $j = 1, \ldots, n$. Under the latent DAG assumption, the ODE system (3) is $\boldsymbol{\eta}$-identifiable from discrete observations $\boldsymbol{x}_1, \ldots, \boldsymbol{x}_n$, if and only if assumption **C1** is satisfied.*

> **C1:** *there exists $(d+p)$ $\boldsymbol{y}_j$'s with indices denoting as $\{j_1, j_2, \ldots, j_{d+p}\} \subseteq \{1, 2, \ldots, n\}$, such that the set of vectors $\{\boldsymbol{y}_{j_1}, \boldsymbol{y}_{j_2}, \ldots, \boldsymbol{y}_{j_{d+p}}\}$ is linearly independent.*

The proof of Theorem 4.2 can be found in Appendix D.3. Condition **C1** is both sufficient and necessary. This theorem states that as long as there are $d + p$ observations $\boldsymbol{x}_j$'s such that the corresponding augmented new observations $\boldsymbol{y}_j$'s are linearly independent, the ODE system (3) is $\boldsymbol{\eta}$-identifiable from these discrete observations. Under the latent DAG assumption, we can transfer the ODE system (3), which includes hidden confounders, into a $(d+p)$-dimensional fully observable ODE system (1) through the augmented state $\boldsymbol{y}(t)$. Condition **C1** indicates that our observations span the entire $(d+p)$-dimensional state space, thus rendering the system identifiable.

Both Definition 4.1 and Definition 4.2 define the identifiability of the ODE system (3) to some extent of the unknown parameters. In other words, given the available observations, under Definition 4.1 and Definition 4.2, one can only identify the values of $(\boldsymbol{x}_0, A, B\boldsymbol{z}_0, BG\boldsymbol{z}_0, \ldots, BG^{p-1}\boldsymbol{z}_0)$, but not the values of $(\boldsymbol{z}_0, B, G)$. Based on Equation (7), this level of identifiability is sufficient to identify the causal relationships between observable variables $\boldsymbol{x}$, enabling safe intervention on the observable variables with reliable causal inferences. However, in scenarios where practitioners can intervene in the latent variables and require inferring the causal effects of the intervened system, identifying the matrices $B$ and $G$ becomes essential for reliable causal references. For instance, in chemical kinetics, where the evolution of chemical concentrations over time can often be modeled by an ODE system [8, 12], some chemicals may not be measurable during the reaction, rendering them latent variables. Nonetheless, practitioners can intervene in these latent variables by setting specific initial concentrations. Therefore, we provide an identifiability analysis of the linear ODE system (3) when practitioners can control the initial condition of the latent variables: $\boldsymbol{z}_0$.

### 4.3 Identifiability condition from $p$ controllable whole trajectories

Assuming the initial condition of the latent variables $\boldsymbol{z}_0$ is controllable, which means that the values of $\boldsymbol{z}_0$ can be treated as given values, we denote it as $\boldsymbol{z}_0^*$. In the following, we provide the identifiability condition of the ODE system (3) when we are given $p$ initial conditions $\boldsymbol{z}_0^*$, denoting as $\boldsymbol{z}_0^{*i}$. We first present the definition.

**Definition 4.3.** *Given $\boldsymbol{z}_0^{*i} \in \mathbb{R}^p$ for $i = 1, \ldots, p$, for $\boldsymbol{x}_0 \in \mathbb{R}^d, A \in \mathbb{R}^{d \times d}, B \in \mathbb{R}^{d \times p}$ and $G \in \mathbb{R}^{p \times p}$, under the latent DAG assumption, for all $\boldsymbol{x}_0' \in \mathbb{R}^d$, all $A' \in \mathbb{R}^{d \times d}$, all $B' \in \mathbb{R}^{d \times p}$, and all $G' \in \mathbb{R}^{p \times p}$, we denote $\boldsymbol{\eta}_i := (\boldsymbol{x}_0, \boldsymbol{z}_0^{*i}, A, B, G)$ and $\boldsymbol{\eta}_i' := (\boldsymbol{x}_0', \boldsymbol{z}_0^{*i}, A', B', G')$, we say the ODE system (3) is $\{\boldsymbol{\eta}_i\}_1^p$-identifiable: if $(\boldsymbol{x}_0, A, B, G) \neq (\boldsymbol{x}', A', B', G')$, it holds that $\exists i$ such that $\boldsymbol{x}(\cdot; \boldsymbol{\eta}_i) \neq \boldsymbol{x}(\cdot; \boldsymbol{\eta}_i')$.*

Definition 4.3 defines the identifiability of the ODE system (3) from $p$ whole trajectories $\boldsymbol{x}(\cdot; \boldsymbol{\eta}_i)$ with $i = 1, \ldots, p$, and under this definition, matrix $B$ and $G$ are also identifiable. Based on this definition, we provide the identifiability condition.

**Theorem 4.3.** *Given $\boldsymbol{z}_0^{*i} \in \mathbb{R}^p$ for $i = 1, \ldots, p$, for $\boldsymbol{x}_0 \in \mathbb{R}^d, A \in \mathbb{R}^{d \times d}, B \in \mathbb{R}^{d \times p}$ and $G \in \mathbb{R}^{p \times p}$, under the latent DAG assumption, the ODE system (3) is $\{\boldsymbol{\eta}_i\}_1^p$-identifiable if assumptions $\boldsymbol{B}_2$, $\boldsymbol{B}_3$ and $\boldsymbol{B}_4$ are all satisfied.*

- **B2**: *each $\boldsymbol{z}_0^{*i}$ for $i = 1, \ldots, p$, satisfies assumption **B1**. That is, if we set $\boldsymbol{\gamma}_i = A^p \boldsymbol{x}_0 + \sum_{j=0}^{p-1} A^{p-1-j} BG^j \boldsymbol{z}_0^{*i}$, then the set of vectors $\{\boldsymbol{\gamma}_i, A\boldsymbol{\gamma}_i, \ldots, A^{d-1}\boldsymbol{\gamma}_i\}$ is linearly independent for all $i = 1, \ldots, p$.*

- **B3**: *the set of vectors $\{\boldsymbol{z}_0^{*1}, \boldsymbol{z}_0^{*2}, \ldots, \boldsymbol{z}_0^{*p}\}$ is linearly independent.*

- **B4**: *the matrix composed by vertically stack the matrices $\{B, BG, \ldots, BG^{p-1}\}$ has rank $p$.*

The proof of Theorem 4.3 can be found in Appendix D.4. Assumption **B2** ensures that the ODE system (3) is $\boldsymbol{\eta}_i$-identifiable for all $i = 1, \ldots, p$. Consequently, $(\boldsymbol{x}_0, A, B\boldsymbol{z}_0^{*i}, BG\boldsymbol{z}_0^{*i}, \ldots, BG^{p-1}\boldsymbol{z}_0^{*i})$ for all $i = 1, \ldots, p$ is identifiable. Then, under assumption **B3**, the identifiability of matrix $B$ is established. To identify matrix $G$, assumption **B4** is required. While the ability to control the initial condition of the latent variables may appear strict, it is a reasonable assumption in our context. This is because identifying matrices $B$ and $G$ is necessary only when practitioners can intervene in the latent variables, thereby allowing control over their initial conditions. An alternative approach to identifying $B$ and $G$ involves intervening in the initial condition of each latent variable $z_i$ independently, rather than controlling the initial condition of all latent variables $\boldsymbol{z}$ simultaneously. This method draws inspiration from the "genetic single-node intervention" proposed by [32], where one can intervene at each latent node individually. Further details of this method can be found in Appendix F.

### 4.4 Identifiability condition from discrete observations sampled from $p$ controllable trajectories

We also extend the identifiability analysis of the ODE system (3) to cases where only discrete observations from $p$ controllable trajectories are available.

**Definition 4.4.** *Given $\boldsymbol{z}_0^{*i} \in \mathbb{R}^p$ for $i = 1, \ldots, p$, for $\boldsymbol{x}_0 \in \mathbb{R}^d, A \in \mathbb{R}^{d \times d}, B \in \mathbb{R}^{d \times p}$ and $G \in \mathbb{R}^{p \times p}$. For any $n \geqslant 1$, let $t_j, j = 1, \ldots, n$ be any $n$ time points and $\boldsymbol{x}_{ij} := \boldsymbol{x}(t_j; \boldsymbol{\eta}_i)$ be the*

*error-free observation of the trajectory $\boldsymbol{x}(\cdot\,;\boldsymbol{\eta}_i)$ at time $t_j$. Under the latent DAG assumption, we say the ODE system* (3) *is $\{\boldsymbol{\eta}_i\}_1^p$-identifiable from $\boldsymbol{x}_{i1},\dots,\boldsymbol{x}_{in}$, $i=1,\dots,p$, if for all $\boldsymbol{x}_0' \in \mathbb{R}^d$, all $A' \in \mathbb{R}^{d\times d}$, all $B' \in \mathbb{R}^{d\times p}$, and all $G' \in \mathbb{R}^{p\times p}$ with $(\boldsymbol{x}_0, A, B, G) \neq (\boldsymbol{x}_0', A', B', G')$, it holds that $\exists i \in \{1,\dots,p\}$ and $j \in \{1,\dots,n\}$ such that $\boldsymbol{x}(t_j;\boldsymbol{\eta}_i) \neq \boldsymbol{x}(t_j;\boldsymbol{\eta}_i')$.*

Based on Definition 4.4 we present the identifiability condition.

**Theorem 4.4.** *Given $\boldsymbol{z}_0^{*i} \in \mathbb{R}^p$ for $i=1,\dots,p$, for $\boldsymbol{x}_0 \in \mathbb{R}^d, A \in \mathbb{R}^{d\times d}, B \in \mathbb{R}^{d\times p}$ and $G \in \mathbb{R}^{p\times p}$. We define new observation $\boldsymbol{y}_{ij} := [\boldsymbol{x}_{ij}^T, 1, t_j, t_j^2, \dots, t_j^{p-1}]^T \in \mathbb{R}^{d+p}$, for $i=1,\dots,p$ and $j=1,\dots,n$. Under the latent DAG assumption, the ODE system* (3) *is $\{\boldsymbol{\eta}_i\}_1^p$-identifiable from discrete observations $\boldsymbol{x}_{i1},\dots,\boldsymbol{x}_{in}$, $i=1,\dots,p$, if assumptions **C2**, **B3** and **B4** are all satisfied.*

> **C2**: *for each $i \in \{1,\dots,p\}$ there exists $(d+p)$ $\boldsymbol{y}_{ij}$'s with indexes denoting as $\{j_{i1}, j_{i2}, \dots, j_{i,d+p}\} \subseteq \{1,2,\dots,n\}$, such that the set of vectors $\{\boldsymbol{y}_{ij_{i1}}, \boldsymbol{y}_{ij_{i2}}, \dots, \boldsymbol{y}_{ij_{i,d+p}}\}$ is linearly independent.*

The proof of Theorem 4.4 can be found in Appendix D.5. Assumption **C2** ensures that the ODE system (3) is $\boldsymbol{\eta}_i$-identifiable from discrete observations $\boldsymbol{x}_{i1},\dots,\boldsymbol{x}_{in}$ for all $i=1,\dots,p$. As in Subsection 4.3, under assumptions **B3** and **B4**, the matrices $B$ and $G$ are also identifiable.

# 5 Simulations

To evaluate the validity of the identifiability conditions established in Section 3 and 4, we present the results of simulations. As previously indicated, the ODE system (3) can be treated as a more intricate version of the ODE system (2); hence, our simulation experiments are centered on the former.

**Simulation design.** We conduct four sets of simulations, which include one identifiable case and one unidentifiable case for both the $\boldsymbol{\eta}$-identifiable check and the $\{\boldsymbol{\eta}_i\}_1^p$-identifiable check. The dimensions of both observable variables, $d$, and latent variables, $p$, are set to 3. The true underlying parameters of the systems are provided below. Observations are simulated from the true ODE systems for each case, with $n$ equally-spaced observations generated from the time interval $[0, 1]$ for each trajectory, and we only keep the values of the observable variables $\boldsymbol{x}$.

$$A = \begin{bmatrix} 2 & -2 & 1 \\ 1 & 1 & -1 \\ 1 & 0 & 2 \end{bmatrix}, \quad B = \begin{bmatrix} -2 & -2 & 2 \\ 0 & -1 & -2 \\ -1 & -1 & -2 \end{bmatrix}, \quad G = \begin{bmatrix} 0 & 2 & 1 \\ 0 & 0 & -2 \\ 0 & 0 & 0 \end{bmatrix}, \quad A' = \begin{bmatrix} 1 & 0 & 0 \\ 0 & 1 & 0 \\ 0 & 0 & 1 \end{bmatrix},$$

$$\boldsymbol{x}_0 = \begin{bmatrix} -1 \\ 1 \\ 1 \end{bmatrix}, \quad \boldsymbol{z}_0 = \begin{bmatrix} 1 \\ -2 \\ -1 \end{bmatrix}, \quad \boldsymbol{z}_0^{*1} = \begin{bmatrix} 1 \\ 0 \\ 0 \end{bmatrix}, \quad \boldsymbol{z}_0^{*2} = \begin{bmatrix} 0 \\ 1 \\ 0 \end{bmatrix}, \quad \boldsymbol{z}_0^{*3} = \begin{bmatrix} 0 \\ 0 \\ 1 \end{bmatrix}.$$

$\boldsymbol{\eta}$-identifiable: $\boldsymbol{\eta} = (\boldsymbol{x}_0, \boldsymbol{z}_0, A, B, G)$, unidentifiable: $\boldsymbol{\eta} = (\boldsymbol{x}_0, \boldsymbol{z}_0, A', B, G)$.

$\{\boldsymbol{\eta}_i\}_1^p$-identifiable: $\boldsymbol{\eta}_i = (\boldsymbol{x}_0, \boldsymbol{z}_0^{*i}, A, B, G)$, unidentifiable: $\boldsymbol{\eta}_i = (\boldsymbol{x}_0, \boldsymbol{z}_0^{*i}, A', B, G), i=1, 2, 3$.

**Parameter estimation.** The Nonlinear Least Squares (NLS) method is employed for parameter estimation, a widely used technique for estimating parameters in nonlinear regression models, including ODEs [7, 21, 46]. The *"least_squares"* function from the *"scipy.optimize"* Python module, with default hyperparameter settings, is utilized for implementation. Given that the NLS loss function for our simulation is non-convex, parameter initialization is performed near the true values to promote convergence to the global minimum. Specifically, for the $\boldsymbol{\eta}$-(un)identifiable cases, initial parameter values are set to the true parameters plus a random value drawn from a uniform distribution $U(-0.1, 0.1)$ for each replication. For $\{\boldsymbol{\eta}_i\}_1^p$-(un)identifiable cases, initial parameter values are set to the true values plus a random value from $U(-0.3, 0.3)$.

**Evaluation metric.** Mean Squared Error (MSE) is adopted as the metric to assess the accuracy of the parameter estimator. To ensure the reliability of the estimation results, 100 independent random replications are run for each configuration, and we report the mean and variance of the squared error.

**Results analysis.** Table 1 and Table 2 present the simulation results for the $\boldsymbol{\eta}$-(un)identifiable cases and the $\{\boldsymbol{\eta}_i\}_1^p$-(un)identifiable cases, respectively. According to Definition 4.1 and Definition 4.3, for the $\boldsymbol{\eta}$-(un)identifiable cases, the identifiability of $(\boldsymbol{x}_0, A, B\boldsymbol{z}_0, BG\boldsymbol{z}_0, BG^2\boldsymbol{z}_0)$ needs to be checked, while for the $\{\boldsymbol{\eta}_i\}_1^p$-(un)identifiable cases, we need to check the identifiability of $(\boldsymbol{x}_0, A, B, G)$.

Since $x_0$ is consistently identifiable (with MSE less than 1.00E-10) across all (un)identifiable cases, its results are not presented.

In both Tables, for identifiable cases, as the number of samples $n$ increases, the MSEs for all parameters of interest decrease and approach zero. However, in the unidenfiable cases, where the identifiability condition **B1/B2** stated in Theorem 4.1/4.3 is unmet, the MSEs for certain parameters remain high irrespective of sample size. These results offer strong empirical support for the validity of the identifiability conditions outlined in Theorem 4.1 and Theorem 4.3. It is noteworthy that in the $\{\boldsymbol{\eta}_i\}_1^p$ case, where observations are sampled from $p = 3$ controllable trajectories, remarkably accurate parameter estimates can be obtained even with a limited number of samples.

Table 1: MSEs of the $\boldsymbol{\eta}$-(un)identifiable cases of the ODE (3)

| $n$ | Identifiable | | | | Unidentifiable | | | |
|---|---|---|---|---|---|---|---|---|
| | $A$ | $Bz_0$ | $BGz_0$ | $BG^2z_0$ | $A$ | $Bz_0$ | $BGz_0$ | $BG^2z_0$ |
| 10 | 6.00E-05 ($\pm$5.40E-08) | 0.0004 ($\pm$3.45E-06) | 0.0044 ($\pm$0.0004) | 0.0007 ($\pm$3.91E-06) | 0.0994 ($\pm$0.0157) | 0.0494 ($\pm$0.1243) | 0.9185 ($\pm$8.3148) | 0.6482 ($\pm$1.4306) |
| 100 | 4.15E-05 ($\pm$1.62E-08) | 0.0003 ($\pm$8.52E-07) | 0.0029 ($\pm$9.42E-05) | 0.0005 ($\pm$2.90E-06) | 0.0372 ($\pm$0.0032) | 0.0174 ($\pm$0.0087) | 0.3517 ($\pm$0.3460) | 0.5767 ($\pm$1.4055) |
| 500 | 2.65E-05 ($\pm$8.71E-09) | 0.0002 ($\pm$4.38E-07) | 0.0019 ($\pm$4.84E-05) | 0.0002 ($\pm$8.38E-07) | 0.0461 ($\pm$0.0099) | 0.1071 ($\pm$0.1768) | 0.5783 ($\pm$2.5747) | 0.3648 ($\pm$0.4507) |

Table 2: MSEs of the $\{\boldsymbol{\eta}_i\}_1^p$-(un)identifiable cases of the ODE (3)

| $n$ | Identifiable | | | Unidentifiable | | |
|---|---|---|---|---|---|---|
| | $A$ | $B$ | $G$ | $A$ | $B$ | $G$ |
| 10 | 5.83E-22 ($\pm$7.41E-42) | 2.85E-21 ($\pm$2.75E-40) | 2.27E-21 ($\pm$5.69E-41) | 0.6349 ($\pm$0.7464) | 0.1913 ($\pm$0.0686) | 0.0044 ($\pm$0.0011) |
| 30 | 1.50E-22 ($\pm$3.23E-43) | 7.80E-22 ($\pm$1.14E-41) | 5.76E-22 ($\pm$5.28E-42) | 0.6169 ($\pm$0.7194) | 0.1850 ($\pm$0.0657) | 0.0045 ($\pm$0.0007) |
| 50 | 5.16E-23 ($\pm$6.20E-44) | 3.01E-22 ($\pm$3.27E-42) | 2.39E-22 ($\pm$8.46E-43) | 0.5876 ($\pm$0.6895) | 0.1761 ($\pm$0.0627) | 0.0045 ($\pm$0.0008) |

For the $\boldsymbol{\eta}$-(un)identifiable cases, assumption **C1** stated in Theorem 4.2 holds true for all values of $n$ in the identifiable cases, while it is violated across all $n$ in the unidentifiable cases. In the $\{\boldsymbol{\eta}_i\}_1^p$-(un)identifiable cases, condition **C2** stated in Theorem 4.4 is satisfied for all values of $n$ in the identifiable cases, but is found to be violated for all values of $n$ in the unidentifiable cases. These findings provide strong empirical evidence supporting the validity of the identifiability conditions proposed in Theorem 4.2 and Theorem 4.4.

In Appendix G, we present additional simulation results for higher-dimensional cases, along with simulations that incorporate a variety of ground-truth parameter configurations. These results consistently affirm the validity of our proposed identifiability conditions. For further details, please refer to Appendix G.

## 6 Related work

**Identifiability analysis of linear ODE systems.** Within control theory, extensive research has been conducted on the identifiability analysis of linear dynamical systems governed by ODEs [5, 14, 15, 17]. In the applied mathematics area, Stanhope et al. [34] and Qiu et al. [28] have systematically investigated the identifiability of linear ODE systems based on a single trajectory. Furthermore, Wang et al. [42] have extended these findings to scenarios where only discrete observations sampled

from a single trajectory are available. However, existing studies primarily concentrate on linear ODE systems with fully observable variables. To the best of our knowledge, our work represents the inaugural endeavor to systematically analyze the identifiability of linear ODE systems in the presence of hidden confounders.

**Connection between causality and differential equations.** Differential equations provide a natural framework for understanding causality within dynamic systems, particularly in the context of continuous-time processes [1, 31]. Consequently, significant efforts have been directed towards establishing a theoretical link between causality and differential equations. In the deterministic case, Mooij et al. [25] and Rubenstein et al. [30] have established a mathematical connection between ODEs and Structural Causal Models (SCMs). Wang et al. [42] have proposed a method for inferring the causal structure of linear ODEs. In the domain of neural ODEs, Aliee et al. [2, 3] have applied various regularization techniques to enhance the recovery of the causal relationships. Turning to the stochastic case, Hansen et al. [18] and Wang et al. [41] have proposed causal interpretations and identifiability analysis of Stochastic Differential Equations (SDEs). Additionally, Bellot et al. [6] have introduced a method for consistently discovering the causal structure of SDE systems using penalized neural ODEs. These works aim to establish a theoretical connection between causality and differential equations in various ways. Our contribution to this scholarly landscape lies in the systematic analysis of the identifiability of linear ODEs, particularly in the presence of hidden confounders.

# 7 Conclusion

This paper presents a systematic identifiability analysis of linear ODE systems incorporating hidden confounders. Specifically, we establish a sufficient and necessary identifiability condition for the linear ODE system with independent latent confounders. Additionally, we provide four identifiability conditions for the linear ODE system with causally related latent confounders, wherein the causal structure of the latent confounders adheres to a DAG.

A notable limitation of our work lies in the practical verification of these identifiability conditions, given that the true underlying system parameters are often unavailable in real-world scenarios. However, our study significantly contributes to the understanding of the intrinsic structure of linear ODE systems with hidden confounders. By providing insights into the identifiability aspects, our findings empower practitioners to utilize models that adhere to the proposed conditions (e.g., through constrained parameter estimation) for learning from real-world data while ensuring identifiability.

# Acknowledgements

YW was supported by the Australian Government Research Training Program (RTP) Scholarship from the University of Melbourne. BH was supported by NSF DMS-2428058. XG was supported by ARC DE210101352. MG was supported by ARC DE210101624 and ARC DP240102088.

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

# Appendix for "Identifiability Analysis of Linear ODE Systems with Hidden Confounders"

## A  Summary of notations and proposed identifiability conditions

Table 3: Summary of notations

| Notation | Description |
|---|---|
| $\boldsymbol{x}/\boldsymbol{z}$ | observable/latent variables |
| $x_i/z_i$ | the $i$-th observable/latent variable |
| $t$ | time |
| $t_j$ | the $j$-th time point |
| $\boldsymbol{x}(t)/\boldsymbol{z}(t)$ | state of observable/latent variable at time $t$ |
| $\boldsymbol{x}_j$ | $\boldsymbol{x}(t_j)$, observable state at time $t_j$ |
| $\boldsymbol{x}_0/\boldsymbol{z}_0$ | initial condition of observable/latent variable |
| $\dot{\boldsymbol{x}}(t)$ | first derivative of $\boldsymbol{x}(t)$ w.r.t. time $t$ |
| $d$ | dimension of observable variables |
| $p$ | dimension of latent variables |
| $A, B, G$ | constant parameter matrices defined in Eq.(2) and (3) |
| $\boldsymbol{f}(t)$ | Function of time $t$ defined in Eq.(2) |
| $\boldsymbol{v}_k$ | constant parameter vector defined in Eq.(4) |
| $\{\boldsymbol{v}_k\}_0^r$ | all the $\boldsymbol{v}_k$'s for $k = 0, \ldots, r$ |
| $\boldsymbol{\theta}$ | $:= (\boldsymbol{x}_0, \boldsymbol{z}_0, A, B, \{\boldsymbol{v}_k\}_0^r)$, the system parameter of ODE system (2) |
| $\boldsymbol{\beta}$ | a vector defined in Thm.3.1 **A1** |
| $\boldsymbol{y}(t)$ | augmented state |
| $\boldsymbol{y}_0$ | initial condition of augmented variable |
| $\boldsymbol{\eta}$ | $:= (\boldsymbol{x}_0, \boldsymbol{z}_0, A, B, G)$, the system parameter of ODE system (3) |
| $\boldsymbol{\gamma}$ | a vector defined in Thm.4.1 **B1** |
| $\boldsymbol{z}_0^*$ | given initial condition of latent variable |
| $\boldsymbol{z}_0^{*i}$ | the $i$-th given initial condition of latent variable |
| $\boldsymbol{\eta}_i$ | $:= (\boldsymbol{x}_0, \boldsymbol{z}_0^{*i}, A, B, G)$, the system parameter of ODE system (3) |
| $\boldsymbol{\gamma}_i$ | a vector defined in Thm 4.3 **B2** |
| $\boldsymbol{x}_{ij}$ | $:= \boldsymbol{x}(t_j; \boldsymbol{\eta}_i)$, observable state of ODE system (3) with parameter $\boldsymbol{\eta}_i$ at time $t_j$ |
| $\boldsymbol{y}_{ij}$ | augmented state of $\boldsymbol{x}_{ij}$ at time $t_j$ |
| $A', \boldsymbol{x}_0', \ldots$ | the alternative counterpart corresponding to $A, \boldsymbol{x}_0, \ldots$ |

Table 4: Summary of proposed identifiability conditions

| ODEs | Conds. | # Traj. | Obs. | Def./Thm. | Necessity |
|---|---|---|---|---|---|
| Eq.(2)+(4) | A1 | 1 | continuous | 3.1 | Yes |
| Eq.(3) | latent DAG, B1 | 1 | continuous | 4.1 | Yes |
| Eq.(3) | latent DAG, C1 | 1 | discrete | 4.2 | Yes |
| Eq.(3) | latent DAG, B2, B3, B4 | $p$ | continuous | 4.3 | No |
| Eq.(3) | latent DAG, C2, B3, B4 | $p$ | discrete | 4.4 | No |

# B  Real world examples

In this section, we present two real-world examples that correspond to the ODE systems (2) and (3). These examples initially assume fully observable systems, with latent variables introduced by us based on prior experience or established physical laws.

## B.1  Damped harmonic oscillators model

Consider a one-dimensional system comprising $D$ point masses $m_i$ for $i = 1, \ldots, D$ with positions $Q_i(t) \in \mathbb{R}$ and momenta $P_i(t) \in \mathbb{R}$. These masses are interconnected by springs characterized by spring constants $k_i$ and equilibrium lengths $l_i$, and each mass is subject to friction with coefficient $b_i$. The system's boundary conditions are fixed at $Q_0(t) = 0$ and $Q_{D+1}(t) = L$.

The dynamics of this system are described by the following linear ODE system [25]:

$$\dot{P}_i(t) = k_i(Q_{i+1}(t) - Q_i(t) - l_i) - k_{i-1}(Q_i(t) - Q_{i-1}(t) - l_{i-1}) - b_i P_i(t)/m_i$$
$$\dot{Q}_i(t) = P_i(t)/m_i \tag{8}$$

where $Q_0(t) = 0$ and $Q_{D+1}(t) = L$ represent the fixed boundary conditions. External forces $F_j(t)$ (e.g., wind force or a varying magnetic field) may influence the entire system of coupled oscillators. These external forces can be modeled here as latent variables with constant derivatives. Consequently, the system can be reformulated as follows:

$$\dot{P}_i(t) = k_i(Q_{i+1}(t) - Q_i(t) - l_i) - k_{i-1}(Q_i(t) - Q_{i-1}(t) - l_{i-1}) - b_i P_i(t)/m_i + \sum_j \alpha_{ij} F_j(t)$$

$$\dot{Q}_i(t) = P_i(t)/m_i$$
$$\dot{F}_j(t) = c_j$$

$$\tag{9}$$

where $\alpha_{ij}$ is a constant determining the effect of the external force $F_j(t)$ on the $i$-th mass, and $c_j$ is the constant rate of change of the external force $F_j(t)$. This model aligns with our ODE system (2), and an illustrative causal structure for this model is provided in Figure 3.

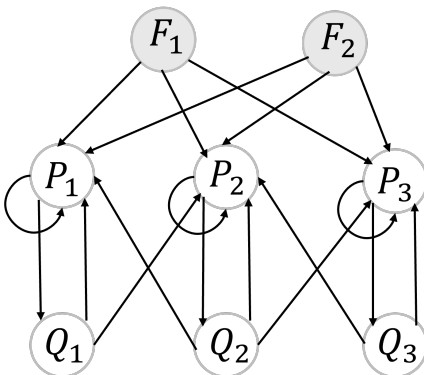

Figure 3: Example causal structure of the damped harmonic oscillators model with 3 oscillators and 2 latent variables.

In regions with predictable wind patterns, such as during monsoon seasons or in controlled experimental settings, wind force can be approximated with a constant rate, making this an ideal context for modeling external forces with constant derivatives. Furthermore, constant forces or those represented as polynomial functions of time align well with our ODE system structure. For instance, a uniform magnetic field acting on the system would produce a constant force. These examples demonstrate that various latent factors can effectively fit within our ODE structure.

## B.2 Population model

The growth of a population $P(t)$ can be described by a linear ODE [4]:

$$\dot{P}(t) = aP(t),$$

where $a$ is a constant representing the population growth rate. This system may also be influenced by latent variables $L_i$, such as environmental factors or food supply. By incorporating these latent influences, the system can be expressed as:

$$\dot{P}(t) = aP(t) + bL_1(t) + cL_2(t)$$
$$\dot{L}_1(t) = lL_2(t)$$
$$\dot{L}_2(t) = m$$

where $a, b, c, l$ and $m$ are constants. Here, $L_1(t)$ represents the food supply, which is influenced by the environmental factor $L_2(t)$. $L_2(t)$ corresponds to an environmental factor, such as temperature or pollution level, that changes steadily over time. This model aligns well with our ODE system (3), and an illustrative causal structure for this model is provided in Figure 4.

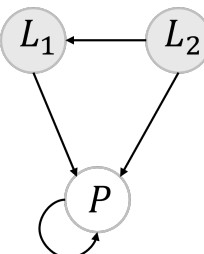

Figure 4: Causal structure of the population model.

An example of an environmental factor changing at a constant rate is pollution from an industrial plant that continuously releases a fixed amount of pollutants, or from a wastewater treatment plant that discharges a specified amount of treated wastewater into a river on an hourly basis.

# C    An example of an unidentifiable case of the linear ODE system (3)

Recall that the parameters of the ODE system (3) are:

$$\boldsymbol{x}_0 = \begin{bmatrix} 1 \\ 1 \end{bmatrix}, \quad \boldsymbol{z}_0 = \begin{bmatrix} 1 \\ 1 \end{bmatrix}, \quad B = \begin{bmatrix} 1 & 1 \\ 1 & 1 \end{bmatrix}, \quad G = \begin{bmatrix} 0 & 1 \\ 0 & 0 \end{bmatrix},$$

$$A = \begin{bmatrix} 1 & 0 \\ 0 & 1 \end{bmatrix}, \quad A' = \begin{bmatrix} 0 & 1 \\ 1 & 0 \end{bmatrix}, \quad M = \begin{bmatrix} A & B \\ \mathbf{0} & G \end{bmatrix}, \quad M' = \begin{bmatrix} A' & B \\ \mathbf{0} & G \end{bmatrix}.$$

We first calculate the solution of $\boldsymbol{z}(t)$,

$$\boldsymbol{z}(t) = e^{Gt}\boldsymbol{z}_0$$

$$= \sum_{k=0}^{\infty} \frac{G^k \boldsymbol{z}_0}{k!} t^k = \sum_{k=0}^{1} \frac{G^k \boldsymbol{z}_0}{k!} t^k = \begin{bmatrix} 1+t \\ 1 \end{bmatrix}$$

We intervene $x_1(t) = 1$, then under matrix $M$:

$$\dot{x}_2(t) = x_2(t) + z_1(t) + z_2(t)$$
$$= x_2(t) + t + 2.$$

To solve this differential equation, we rewrite it in the standard linear form and multiply both sides by the integrating factor $e^{-t}$,

$$e^{-t}\dot{x}_2(t) - e^{-t}x_2(t) = (t+2)e^{-t}.$$

The left-hand side of this equation is the derivative of $e^{-t}x_2(t)$:

$$\frac{d}{dt}(e^{-t}x_2(t)) = (t+2)e^{-t}.$$

Next, integrate both sides w.r.t. $t$:

$$\int \frac{d}{dt}(e^{-t}x_2(t))dt = \int (t+2)e^{-t}dt.$$

The left-hand side integrates to:

$$e^{-t}x_2(t).$$

Next, we use integration by parts to find the integral on the right-hand side:

$$\int (t+2)e^{-t}dt = -(t+2)e^{-t} - \int -e^{-t}dt$$
$$= -(t+2)e^{-t} - e^{-t}$$
$$= -(t+3)e^{-t}.$$

Thus:

$$e^{-t}x_2(t) = -(t+3)e^{-t} + C,$$

where $C$ is the constant of the integration.

Multiplying both sides by $e^t$ to solve for $x_2(t)$:

$$x_2(t) = -t - 3 + Ce^t.$$

Now, use the initial condition $x_2(0) = 1$, we get

$$C = 4.$$

Therefore,

$$x_2(t) = 4e^t - t - 3.$$

Whereas under matrix $M'$:

$$\dot{x}_2(t) = x_1(t) + z_1(t) + z_2(t)$$
$$= t + 3.$$

Simple calculations show that

$$\boldsymbol{x}_2(t) = t^2/2 + 3t + 1.$$

# D   Detailed proofs

## D.1   Proof of Theorem 3.1

*Proof.* Recall that the first derivative of $x(t)$ can be expressed as:

$$\dot{x}(t) = Ax(t) + Bz(t)$$
$$= Ax(t) + \sum_{k=0}^{r} \frac{Bv_k}{k+1}t^{k+1} + Bz_0.$$

Set

$$y(t) = \begin{bmatrix} x(t) \\ 1 \\ t \\ t^2 \\ \vdots \\ t^{r+1} \end{bmatrix},$$

we see that $y(t) \in \mathbb{R}^{d+r+2}$, and the first derivative of $y(t)$ w.r.t. time $t$ can be expressed as

$$\dot{y}(t) = \begin{bmatrix} \dot{x}(t) \\ 0 \\ 1 \\ 2t \\ \vdots \\ (r+1)t^r \end{bmatrix}$$

$$= \underbrace{\begin{bmatrix} A & Bz_0 & Bv_0 & \frac{Bv_1}{2} & \cdots & \frac{Bv_{r-1}}{r} & \frac{Bv_r}{r+1} \\ \mathbf{0}_d & 0 & 0 & 0 & \cdots & 0 & 0 \\ \mathbf{0}_d & 1 & 0 & 0 & \cdots & 0 & 0 \\ \mathbf{0}_d & 0 & 2 & 0 & \cdots & 0 & 0 \\ \vdots & \vdots & \vdots & \vdots & \ddots & \vdots & \vdots \\ \mathbf{0}_d & 0 & 0 & 0 & \cdots & r+1 & 0 \end{bmatrix}}_{\text{denoted as } F} \underbrace{\begin{bmatrix} x(t) \\ 1 \\ t \\ t^2 \\ \vdots \\ t^{r+1} \end{bmatrix}}_{y(t)},$$

where $\mathbf{0}_d$ denotes a $d$ dimensional zero row vector. Obviously,

$$y(0) = [x_0^T, 1, 0, 0, \ldots, 0]^\top,$$

we denote it as $y_0$. Therefore, $y(t)$ follows a homogeneous linear ODE system that can be expressed as:

$$\begin{aligned} \dot{y}(t) &= Fy(t), \\ y(0) &= y_0, \end{aligned} \tag{10}$$

where $F \in \mathbb{R}^{(d+r+2)\times(d+r+2)}$. Worth noting that all state variables in the ODE system (10) are observable. Then according to Lemma 2.1, the identifiability of the dynamical system described by the ODE system (10) is contingent upon the linear independence of the vectors $\{y_0, Fy_0, F^2y_0, \ldots, F^{d+r+1}y_0\}$. Specifically, the system is $(y_0, F)$-identifiable if and only if this set of vectors is linearly independent, indicating that the matrix formed by these vectors, denoted by $H$, has a rank of $d+r+2$. In the following, we will elucidate that if and only assumption **A1** is satisfied, the rank of this matrix $H$ equals $d+r+2$.

Some calculations show that,

$$
F^k \boldsymbol{y}_0 = \begin{bmatrix} A^{k-1}(A\boldsymbol{x}_0 + B\boldsymbol{z}_0) + \sum_{j=0}^{k-2} j! A^{k-2-j} B\boldsymbol{v}_j \\ 0 \\ \vdots \\ 0 \\ k! \\ 0 \\ \vdots \\ 0 \end{bmatrix} \quad \text{for } k = 1, 2, \ldots, r+1, \qquad (11)
$$

where $k!$ is the $(d+k+1)$-th element.

And

$$
F^k \boldsymbol{y}_0 = \begin{bmatrix} A^{k-(r+2)}(A^{r+2}\boldsymbol{x}_0 + A^{r+1}B\boldsymbol{z}_0 + \sum_{j=0}^{r} j! A^{r-j} B\boldsymbol{v}_j) \\ 0 \\ \vdots \\ 0 \end{bmatrix} \quad \text{for } k = r+2, \ldots, r+d+1.
$$

$$(12)$$

According to assumption **A1** in Theorem 3.1,

$$
\boldsymbol{\beta} = A^{r+2}\boldsymbol{x}_0 + A^{r+1}B\boldsymbol{z}_0 + \sum_{j=0}^{r} j! A^{r-j} B\boldsymbol{v}_j \,,
$$

therefore, $F^k \boldsymbol{y}_0$ can also be expressed as

$$
F^k \boldsymbol{y}_0 = \begin{bmatrix} A^{k-(r+2)}\boldsymbol{\beta} \\ 0 \\ \vdots \\ 0 \end{bmatrix} \quad \text{for } k = r+2, \ldots, r+d+1. \qquad (13)
$$

We denote the matrix

$$
\begin{aligned}
H &:= \begin{bmatrix} \boldsymbol{y}_0 & F\boldsymbol{y}_0 & F^2\boldsymbol{y}_0 & \ldots & F^{r+1}\boldsymbol{y}_0 & F^{r+2}\boldsymbol{y}_0 & \ldots & F^{d+r+1}\boldsymbol{y}_0 \end{bmatrix} \\
&:= \begin{bmatrix} H_{11} & H_{12} \\ H_{21} & H_{22} \end{bmatrix}
\end{aligned}
$$

as a block matrix. Then, based on Equations (11) and (13), one obtains that

$$
\begin{aligned}
H_{11} &= \begin{bmatrix} \boldsymbol{x}_0 & A\boldsymbol{x}_0 + B\boldsymbol{z}_0 & A^2\boldsymbol{x}_0 + AB\boldsymbol{z}_0 + B\boldsymbol{v}_0 & \ldots & A^{r+1}\boldsymbol{x}_0 + A^r B\boldsymbol{z}_0 + \sum_{j=0}^{r-1} j! A^{r-1-j} B\boldsymbol{v}_j \end{bmatrix} \\
&\in \mathbb{R}^{d \times (r+2)} \,, \\
H_{12} &= \begin{bmatrix} \boldsymbol{\beta} & A\boldsymbol{\beta} & \ldots & A^{d-1}\boldsymbol{\beta} \end{bmatrix} \\
&\in \mathbb{R}^{d \times d} \,, \\
H_{21} &= \begin{bmatrix} 1 & 0 & 0 & \ldots & 0 \\ 0 & 1 & 0 & \ldots & 0 \\ 0 & 0 & 2! & \ldots & 0 \\ \vdots & \vdots & \vdots & \ddots & \vdots \\ 0 & 0 & 0 & \ldots & (r+1)! \end{bmatrix} \in \mathbb{R}^{(r+2) \times (r+2)} \,, \\
H_{22} &= \mathbf{0}_{(r+2) \times d} \in \mathbb{R}^{(r+2) \times d} \,.
\end{aligned}
$$

Some calculations show that

$$
\text{rank}(H) = \text{rank}(H_{12}) + \text{rank}(H_{21}) \,.
$$

It is apparent that

$$
\text{rank}(H_{21}) = r + 2 \,.
$$

To achieve $\text{rank}(H) = d + r + 2$, the rank of $H_{12}$ must be $d$. The rank of $H_{12}$ equals $d$ if and only if the set of vectors $\{\boldsymbol{\beta}, A\boldsymbol{\beta}, \ldots, A^{d-1}\boldsymbol{\beta}\}$ is linearly independent, that is, assumption **A1** is satisfied.

Now that we have proved that the ODE system (10) is $(\boldsymbol{y}_0, F)$-identifiable if and only if assumption **A1** is satisfied. That is, under assumption **A1**, the trajectory $\boldsymbol{y}(\cdot; \boldsymbol{y}_0, F)$ uniquely determines both $\boldsymbol{y}_0$ and matrix $F$. Consequently, it also uniquely determines $(\boldsymbol{x}_0, A, B\boldsymbol{z}_0, B\boldsymbol{v}_0, \ldots, B\boldsymbol{v}_r)$, thus establishing that the ODE system (2) is $\boldsymbol{\theta}$-identifiable if and only if assumption **A1** is satisfied. $\quad\square$

### D.2 Proof of Theorem 4.1

*Proof.* Recall that the first derivative of $\boldsymbol{x}(t)$ can be expressed as:

$$\dot{\boldsymbol{x}}(t) = A\boldsymbol{x}(t) + B\boldsymbol{z}(t)$$
$$= A\boldsymbol{x}(t) + \sum_{k=0}^{p-1} \frac{BG^k \boldsymbol{z}_0}{k!} t^k \,.$$

Set

$$\boldsymbol{y}(t) = \begin{bmatrix} \boldsymbol{x}(t) \\ 1 \\ t \\ t^2 \\ \vdots \\ t^{p-1} \end{bmatrix},$$

we see that $\boldsymbol{y}(t) \in \mathbb{R}^{d+p}$, and the first derivative of $\boldsymbol{y}(t)$ w.r.t. time $t$ can be expressed as

$$\dot{\boldsymbol{y}}(t) = \begin{bmatrix} \dot{\boldsymbol{x}}(t) \\ 0 \\ 1 \\ 2t \\ \vdots \\ (p-1)t^{p-2} \end{bmatrix}$$

$$= \underbrace{\begin{bmatrix} A & B\boldsymbol{z}_0 & BG\boldsymbol{z}_0 & \frac{BG^2\boldsymbol{z}_0}{2!} & \cdots & \frac{BG^{p-2}\boldsymbol{z}_0}{(p-2)!} & \frac{BG^{p-1}\boldsymbol{z}_0}{(p-1)!} \\ \mathbf{0}_d & 0 & 0 & 0 & \cdots & 0 & 0 \\ \mathbf{0}_d & 1 & 0 & 0 & \cdots & 0 & 0 \\ \mathbf{0}_d & 0 & 2 & 0 & \cdots & 0 & 0 \\ \vdots & \vdots & \vdots & \vdots & \ddots & \vdots & \vdots \\ \mathbf{0}_d & 0 & 0 & 0 & \cdots & p-1 & 0 \end{bmatrix}}_{\text{denoted as } F} \underbrace{\begin{bmatrix} \boldsymbol{x}(t) \\ 1 \\ t \\ t^2 \\ \vdots \\ t^{p-1} \end{bmatrix}}_{\boldsymbol{y}(t)},$$

where $\mathbf{0}_d$ denotes a $d$ dimensional zero row vector. Obviously,

$$\boldsymbol{y}(0) = [\boldsymbol{x}_0^T, 1, 0, 0, \ldots, 0]^\top \,,$$

we denote it as $\boldsymbol{y}_0$. Therefore, $\boldsymbol{y}(t)$ follows a homogeneous linear ODE system that can be expressed as:

$$\begin{aligned} \dot{\boldsymbol{y}}(t) &= F\boldsymbol{y}(t) \,, \\ \boldsymbol{y}(0) &= \boldsymbol{y}_0 \,, \end{aligned} \tag{14}$$

where $F \in \mathbb{R}^{(d+p)\times(d+p)}$. Worth noting that all state variables in the ODE system (14) are observable. Then according to Lemma 2.1, the identifiability of the dynamical system described by the ODE system (14) is contingent upon the linear independence of the vectors $\{\boldsymbol{y}_0, F\boldsymbol{y}_0, F^2\boldsymbol{y}_0, \ldots, F^{d+p-1}\boldsymbol{y}_0\}$. Specifically, the system is $(\boldsymbol{y}_0, F)$-identifiable if and only if this set of vectors is linearly independent, indicating that the matrix formed by these vectors, denoted by $H$, has a rank of $d+p$. In the following, we will elucidate that if and only assumption **B1** is satisfied, the rank of this matrix $H$ equals $d + p$.

Some calculations show that,

$$F^k \boldsymbol{y}_0 = \begin{bmatrix} A^k \boldsymbol{x}_0 + \sum_{j=0}^{k-1} A^{k-1-j} BG^j \boldsymbol{z}_0 \\ 0 \\ \vdots \\ 0 \\ k! \\ 0 \\ \vdots \\ 0 \end{bmatrix} \qquad \text{for } k = 1, 2, \ldots, p-1, \qquad (15)$$

where $k!$ is the $(d+k+1)$-th element.

And

$$F^k \boldsymbol{y}_0 = \begin{bmatrix} A^{k-p}(A^p \boldsymbol{x}_0 + \sum_{j=0}^{p-1} A^{p-1-j} BG^j \boldsymbol{z}_0) \\ 0 \\ \vdots \\ 0 \end{bmatrix} \qquad \text{for } k = p, p+1, \ldots, p+d-1. \qquad (16)$$

According to assumption **B1** in Theorem 3.1,

$$\boldsymbol{\gamma} = A^p \boldsymbol{x}_0 + \sum_{j=0}^{p-1} A^{p-1-j} BG^j \boldsymbol{z}_0 \,,$$

therefore, $F^k \boldsymbol{y}_0$ can also be expressed as

$$F^k \boldsymbol{y}_0 = \begin{bmatrix} A^{k-p} \boldsymbol{\gamma} \\ 0 \\ \vdots \\ 0 \end{bmatrix} \qquad \text{for } k = p, p+1, \ldots, p+d-1. \qquad (17)$$

We denote the matrix

$$\begin{aligned} H &:= \begin{bmatrix} \boldsymbol{y}_0 & F\boldsymbol{y}_0 & F^2\boldsymbol{y}_0 & \cdots & F^{p-1}\boldsymbol{y}_0 & F^p\boldsymbol{y}_0 & \cdots & F^{p+d-1}\boldsymbol{y}_0 \end{bmatrix} \\ &:= \begin{bmatrix} H_{11} & H_{12} \\ H_{21} & H_{22} \end{bmatrix} \end{aligned}$$

as a block matrix. Then, based on Equations (15) and (17), one obtains that

$$\begin{aligned} H_{11} &= \begin{bmatrix} \boldsymbol{x}_0 & A\boldsymbol{x}_0 + B\boldsymbol{z}_0 & A^2\boldsymbol{x}_0 + AB\boldsymbol{z}_0 + BG\boldsymbol{z}_0 & \cdots & A^{p-1}\boldsymbol{x}_0 + \sum_{j=0}^{p-2} A^{p-2-j} BG^j \boldsymbol{z}_0 \end{bmatrix} \\ &\in \mathbb{R}^{d \times p}, \\ H_{12} &= \begin{bmatrix} \boldsymbol{\gamma} & A\boldsymbol{\gamma} & \cdots & A^{d-1}\boldsymbol{\gamma} \end{bmatrix} \\ &\in \mathbb{R}^{d \times d}, \\ H_{21} &= \begin{bmatrix} 1 & 0 & 0 & \cdots & 0 \\ 0 & 1 & 0 & \cdots & 0 \\ 0 & 0 & 2! & \cdots & 0 \\ \vdots & \vdots & \vdots & \ddots & \vdots \\ 0 & 0 & 0 & \cdots & (p-1)! \end{bmatrix} \in \mathbb{R}^{p \times p}, \\ H_{22} &= \boldsymbol{0}_{p \times d} \in \mathbb{R}^{p \times d}. \end{aligned}$$

Some calculations show that

$$\text{rank}(H) = \text{rank}(H_{12}) + \text{rank}(H_{21}).$$

It is apparent that

$$\text{rank}(H_{21}) = p.$$

To achieve $\text{rank}(H) = d + p$, the rank of $H_{12}$ must be $d$. The rank of $H_{12}$ equals $d$ if and only if the set of vectors $\{\boldsymbol{\gamma}, A\boldsymbol{\gamma}, \ldots, A^{d-1}\boldsymbol{\gamma}\}$ is linearly independent, that is, assumption **B1** is satisfied.

Now that we have proved that the ODE system (14) is $(\boldsymbol{y}_0, F)$-identifiable if and only if assumption **B1** is satisfied. That is, under assumption **B1**, the trajectory $\boldsymbol{y}(\cdot; \boldsymbol{y}_0, F)$ uniquely determines both $\boldsymbol{y}_0$ and the matrix $F$. Consequently, it also uniquely determines $(\boldsymbol{x}_0, A, B\boldsymbol{z}_0, BG\boldsymbol{z}_0, \ldots, BG^{p-1}\boldsymbol{z}_0)$, thus establishing that the ODE system (3) is $\boldsymbol{\eta}$-identifiable if and only if assumption **B1** is satisfied. $\qquad\square$

### D.3 Proof of Theorem 4.2

Before providing the main proof, we first present two lemmas we will use for our proof.

**Lemma D.1.** *[34, Theorem 3.4] The ODE system* (1) *is* $(\boldsymbol{x}_0, A)$-*identifiable if and only if the trajectory* $\boldsymbol{x}(\cdot; \boldsymbol{x}_0, A)$ *is not confined to a proper subspace of* $\mathbb{R}^d$.

**Lemma D.2.** *[34, Lemma 6.1] Trajectory* $\boldsymbol{x}(\cdot; \boldsymbol{x}_0, A)$ *is not confined to a proper subspace of* $\mathbb{R}^d$ *if and only if there exists* $t_1, t_2, \ldots, t_d$ *such that* $\boldsymbol{x}_1, \boldsymbol{x}_2, \ldots, \boldsymbol{x}_d$ *are linearly independent.*

*Proof.* In the proof of Theorem 4.1, we demonstrated that the ODE system (3), under latent DAG assumption, can be transformed into a fully observable homogeneous linear ODE system (14). According to Lemma D.1, the ODE system (14) is $(\boldsymbol{y}_0, F)$-identifiable if and only if trajectory $\boldsymbol{y}(\cdot; \boldsymbol{y}_0, F)$ is not confined to a proper subspace of $\mathbb{R}^{d+p}$. Furthermore, based on Lemma D.2, this condition holds if and only if there exists time points $t_1, t_2, \ldots, t_{d+p}$ such that the vectors $\boldsymbol{y}_1, \boldsymbol{y}_2, \ldots, \boldsymbol{y}_{d+p}$ are linearly independent (i.e., assumption **C1**). Therefore, if and only if assumption **C1** is satisfied, the trajectory $\boldsymbol{y}(\cdot; \boldsymbol{y}_0, F)$ is not confined to a proper subspace of $\mathbb{R}^{d+p}$, ensuring that the ODE system (14) is $(\boldsymbol{y}_0, F)$-identifiable. Consequently, the ODE system (3) is $\boldsymbol{\eta}$-identifiable. $\qquad\square$

### D.4 Proof of Theorem 4.3

*Proof.* Under assumption **B2**, since each $\boldsymbol{z}_0^{*i}$ satisfies assumption **B1**, Theorem 4.1 implies that the ODE system (3) is $\boldsymbol{\eta}_i$-identifiable for all $i = 1, \ldots, p$. That is, one can identify

$$(\boldsymbol{x}_0, A, B\boldsymbol{z}_0^{*i}, BG\boldsymbol{z}_0^{*i}, \ldots, BG^{p-1}\boldsymbol{z}_0^{*i})$$

for all $i = 1, \ldots, p$.

Next, we will prove that matrix $B$ is identifiable under assumption **B3**.

Define the matrix

$$S := \begin{bmatrix} B\boldsymbol{z}_0^{*1} & B\boldsymbol{z}_0^{*2} & \ldots & B\boldsymbol{z}_0^{*p} \end{bmatrix},$$

we know that $S \in \mathbb{R}^{d \times p}$, and $S$ is identifiable. The matrix $S$ can also be expressed as:

$$\begin{aligned} S &= B \begin{bmatrix} \boldsymbol{z}_0^{*1} & \boldsymbol{z}_0^{*2} & \ldots & \boldsymbol{z}_0^{*p} \end{bmatrix} \\ &:= BZ, \end{aligned}$$

where under assumption **B3**, the matrix $Z$ is invertible. Therefore,

$$B = SZ^{-1}.$$

Since $Z$ is a known matrix, $B$ is identifiable.

Similarly, we can prove that $BG^j$ for $j = 1, \ldots, p-1$ is also identifiable.

We now show that, under assumption **B4**, the matrix $G$ is identifiable.

Define the matrix

$$W := \begin{bmatrix} B \\ BG \\ \vdots \\ BG^{p-1} \end{bmatrix},$$

we know that $W \in \mathbb{R}^{dp \times p}$, and $W$ is identifiable.

Since $G$ is a $p \times p$ nilpotent matrix, $G^p = \mathbf{0}$, thus $BG^p = \mathbf{0}$. If we define the matrix

$$V := \begin{bmatrix} BG \\ BG^2 \\ \vdots \\ BG^p \end{bmatrix},$$

then $V \in \mathbb{R}^{dp \times p}$, and $V$ is identifiable. The matrix $V$ can also be expressed as:

$$V = \begin{bmatrix} B \\ BG \\ \vdots \\ BG^{p-1} \end{bmatrix} G = WG. \tag{18}$$

Under assumption **B4**, one can find $p$ linearly independent rows in matrix $W$. Denote the matrix composed of these $p$ linearly independent rows as $W_p$, which is invertible. Denote the matrix composed of the corresponding $p$ rows of $V$ as $V_p$, we have

$$V_p = W_p G.$$

Since $W_p$ is invertible, then

$$G = W_p^{-1} V_p.$$

Because both $V_p$ and $W_p$ are identifiable, $G$ is also identifiable. $\qquad\square$

## D.5  Proof of Theorem 4.4

*Proof.* Under assumption **C2**, for each $i \in \{1, \ldots, p\}$, the corresponding observations satisfy assumption **C1**. Based on Theorem 4.2, the ODE system (3) is $\boldsymbol{\eta}_i$-identifiable for all $i = 1, \ldots, p$. This implies that one can identify

$$(\boldsymbol{x}_0, A, B\boldsymbol{z}_0^{*i}, BG\boldsymbol{z}_0^{*i}, \ldots, BG^{p-1}\boldsymbol{z}_0^{*i})$$

for all $i = 1, \ldots, p$.

According to the proof of Theorem 4.3, under assumptions **B3** and **B4**, matrices $B$ and $G$ are also identifiable. $\qquad\square$

# E  Identifiability conditions of the linear ODE system (2) with other *f(t)*

In this section, we provide identifiability conditions for the linear ODE system (2) with $\boldsymbol{f}(t) = \boldsymbol{v}e^t$ and $\boldsymbol{f}(t) = \boldsymbol{v}_1 sin(t) + \boldsymbol{v}_2 cos(t)$. For notational simplicity, we slightly abuse notation by using the same symbols as in Section 3.

## E.1  When *f(t)* follows an exponenial function of time *t*

We define $\boldsymbol{f}(t)$ in the ODE system (2) as:

$$\boldsymbol{f}(t) = \boldsymbol{v}e^t, \quad \boldsymbol{v} \in \mathbb{R}^p.$$

Simple calculations show that

$$\boldsymbol{z}(t) = \boldsymbol{v}e^t + \boldsymbol{z}_0 - \boldsymbol{v}.$$

Thus,

$$\begin{aligned}
\dot{\boldsymbol{x}}(t) &= A\boldsymbol{x}(t) + B\boldsymbol{z}(t) \\
&= A\boldsymbol{x}(t) + B\boldsymbol{v}e^t + B\boldsymbol{z}_0 - B\boldsymbol{v}.
\end{aligned} \tag{19}$$

We denote the unknown parameters of the ODE system (2) with this $\boldsymbol{f}(t)$ as $\boldsymbol{\theta}$, specifically, $\boldsymbol{\theta} := (\boldsymbol{x}_0, \boldsymbol{z}_0, A, B, \boldsymbol{v})$. Let $[\boldsymbol{x}^T(t; \boldsymbol{\theta}), \boldsymbol{z}^T(t; \boldsymbol{\theta})]^T$ denote the solution of the ODE system (2). It is important to note that under our hidden variables setting, only $\boldsymbol{x}(t; \boldsymbol{\theta})$ is observable. Based on Equation (19), we present the following identifiability definition.

**Definition E.1.** *For $\boldsymbol{x}_0 \in \mathbb{R}^d, \boldsymbol{z}_0 \in \mathbb{R}^p, A \in \mathbb{R}^{d \times d}, B \in \mathbb{R}^{d \times p}$ and $\boldsymbol{v} \in \mathbb{R}^p$, for all $\boldsymbol{x}_0' \in \mathbb{R}^d$, all $\boldsymbol{z}_0' \in \mathbb{R}^p$, all $A' \in \mathbb{R}^{d \times d}$, all $B' \in \mathbb{R}^{d \times p}$, and all $\boldsymbol{v}' \in \mathbb{R}^p$, we denote $\boldsymbol{\theta}' := (\boldsymbol{x}_0', \boldsymbol{z}_0', A', B', \boldsymbol{v}')$, we say the ODE system (2) is $\boldsymbol{\theta}$-identifiable: if $(\boldsymbol{x}_0, A, B\boldsymbol{z}_0, B\boldsymbol{v}) \neq (\boldsymbol{x}_0', A', B'\boldsymbol{z}_0', B'\boldsymbol{v}')$, it holds that $\boldsymbol{x}(\cdot; \boldsymbol{\theta}) \neq \boldsymbol{x}(\cdot; \boldsymbol{\theta}').$*

According to Definition E.1, if the ODE system (2) with an exponential $\boldsymbol{f}(t)$ is $\boldsymbol{\theta}$-identifiable, then the trajectory of the system can uniquely determine the values of $(\boldsymbol{x}_0, A, B\boldsymbol{z}_0, B\boldsymbol{v})$. This determination is sufficient to identify the causal relationships between observable variables $\boldsymbol{x}$ as described by Equation (19). Consequently, one can safely intervene in the observable variables of the ODE system and make reliable causal inferences, despite the fact that matrix $B$ cannot be identified under this definition.

**Theorem E.1.** *For $\boldsymbol{x}_0 \in \mathbb{R}^d, \boldsymbol{z}_0 \in \mathbb{R}^p, A \in \mathbb{R}^{d \times d}, B \in \mathbb{R}^{d \times p}$, and $\boldsymbol{v} \in \mathbb{R}^p$, the ODE system (2) is $\boldsymbol{\theta}$-identifiable if and only if assumption $\boldsymbol{D1}$ is satisfied.*

> $\boldsymbol{D1}$ *the set of vectors $\{\boldsymbol{y}_0, F\boldsymbol{y}_0, \dots, F^{d+1}\boldsymbol{y}_0\}$ is linearly independent, where $\boldsymbol{y}_0 = [\boldsymbol{x}_0^T, 1, 1]^T$, and*
>
> $$F = \begin{bmatrix} A & B\boldsymbol{v} & B\boldsymbol{z}_0 - B\boldsymbol{v} \\ \boldsymbol{0}_d & 1 & 0 \\ \boldsymbol{0}_d & 0 & 0 \end{bmatrix},$$
>
> $\boldsymbol{0}_d$ *denotes a $d$ dimensional zero row vector.*

The proof of Theorem E.1 is presented below. Condition $\boldsymbol{D1}$ is both sufficient and necessary, indicating, from a geometric perspective, that the vector $\boldsymbol{y}_0$ is not contained in an $F$-invariant proper subspace of $\mathbb{R}^{d+2}$.

*Proof.* Set

$$\boldsymbol{y}(t) = \begin{bmatrix} \boldsymbol{x}(t) \\ e^t \\ 1 \end{bmatrix},$$

we see that $\boldsymbol{y}(t) \in \mathbb{R}^{d+2}$, and the first derivative of $\boldsymbol{y}(t)$ w.r.t. time $t$ can be expressed as

$$\dot{\boldsymbol{y}}(t) = \begin{bmatrix} \dot{\boldsymbol{x}}(t) \\ e^t \\ 0 \end{bmatrix} = \underbrace{\begin{bmatrix} A & B\boldsymbol{v} & B\boldsymbol{z}_0 - B\boldsymbol{v} \\ \boldsymbol{0}_d & 1 & 0 \\ \boldsymbol{0}_d & 0 & 0 \end{bmatrix}}_{F} \underbrace{\begin{bmatrix} \boldsymbol{x}(t) \\ e^t \\ 1 \end{bmatrix}}_{\boldsymbol{y}(t)},$$

where $\mathbf{0}_d$ denotes a $d$ dimensional zero row vector. Obviously,

$$\boldsymbol{y}(0) = [\boldsymbol{x}_0^T, 1, 1]^T = \boldsymbol{y}_0\,.$$

Therefore, $\boldsymbol{y}(t)$ follows a homogeneous linear ODE system that can be expressed as:

$$\begin{aligned} \dot{\boldsymbol{y}}(t) &= F\boldsymbol{y}(t)\,, \\ \boldsymbol{y}(0) &= \boldsymbol{y}_0\,, \end{aligned} \tag{20}$$

where $F \in \mathbb{R}^{(d+2)\times(d+2)}$. Worth noting that all state variables in the ODE system (20) are observable. Then according to Lemma 2.1, the system (20) is $(\boldsymbol{y}_0, F)$-identifiable if and only if condition **D1** stated in Theorem E.1 is satisfied. That is, under assumption **D1**, the trajectory $\boldsymbol{y}(\cdot; \boldsymbol{y}_0, F)$ uniquely determines both $\boldsymbol{y}_0$ and matrix $F$. Consequently, it also uniquely determines $(\boldsymbol{x}_0, A, B\boldsymbol{z}_0, B\boldsymbol{v})$, thus establishing that the ODE system (2) is $\boldsymbol{\theta}$-identifiable if and only if assumption **D1** is satisfied. $\quad\square$

### E.2   When *f(t)* follows an trigonometric function of time *t*

We define $\boldsymbol{f}(t)$ in the ODE system (2) as:

$$\boldsymbol{f}(t) = \boldsymbol{v}_1 sin(t) + \boldsymbol{v}_2 cos(t)\,, \quad \boldsymbol{v}_1, \boldsymbol{v}_2 \in \mathbb{R}^p\,.$$

Simple calculations show that

$$\boldsymbol{z}(t) = \boldsymbol{v}_2 sin(t) - \boldsymbol{v}_1 cos(t) + \boldsymbol{z}_0 + \boldsymbol{v}_1\,.$$

Thus,

$$\begin{aligned} \dot{\boldsymbol{x}}(t) &= A\boldsymbol{x}(t) + B\boldsymbol{z}(t) \\ &= A\boldsymbol{x}(t) + B\boldsymbol{v}_2 sin(t) - B\boldsymbol{v}_1 cos(t) + B\boldsymbol{z}_0 + B\boldsymbol{v}_1\,. \end{aligned} \tag{21}$$

We denote the unknown parameters of the ODE system (2) with this $\boldsymbol{f}(t)$ as $\boldsymbol{\theta}$, specifically, $\boldsymbol{\theta} := (\boldsymbol{x}_0, \boldsymbol{z}_0, A, B, \boldsymbol{v}_1, \boldsymbol{v}_2)$. Let $[\boldsymbol{x}^T(t; \boldsymbol{\theta}), \boldsymbol{z}^T(t; \boldsymbol{\theta})]^T$ denote the solution of the ODE system (2). It is important to note that under our hidden variables setting, only $\boldsymbol{x}(t; \boldsymbol{\theta})$ is observable. Based on Equation (21), we present the following identifiability definition.

**Definition E.2.** *For $\boldsymbol{x}_0 \in \mathbb{R}^d, \boldsymbol{z}_0 \in \mathbb{R}^p, A \in \mathbb{R}^{d\times d}, B \in \mathbb{R}^{d\times p}$ and $\boldsymbol{v}_1, \boldsymbol{v}_2 \in \mathbb{R}^p$, for all $\boldsymbol{x}_0' \in \mathbb{R}^d$, all $\boldsymbol{z}_0' \in \mathbb{R}^p$, all $A' \in \mathbb{R}^{d\times d}$, all $B' \in \mathbb{R}^{d\times p}$, and all $\boldsymbol{v}_1', \boldsymbol{v}_2' \in \mathbb{R}^p$, we denote $\boldsymbol{\theta}' := (\boldsymbol{x}_0', \boldsymbol{z}_0', A', B', \boldsymbol{v}_1', \boldsymbol{v}_2')$, we say the ODE system (2) is $\boldsymbol{\theta}$-identifiable: if $(\boldsymbol{x}_0, A, B\boldsymbol{z}_0, B\boldsymbol{v}_1, B\boldsymbol{v}_2) \neq (\boldsymbol{x}_0', A', B'\boldsymbol{z}_0', B'\boldsymbol{v}_1', B'\boldsymbol{v}_2')$, it holds that $\boldsymbol{x}(\cdot; \boldsymbol{\theta}) \neq \boldsymbol{x}(\cdot; \boldsymbol{\theta}')$.*

According to Definition E.2, if the ODE system (2) with a trigonometric $\boldsymbol{f}(t)$ is $\boldsymbol{\theta}$-identifiable, then the trajectory of the system can uniquely determine the values of $(\boldsymbol{x}_0, A, B\boldsymbol{z}_0, B\boldsymbol{v}_1, B\boldsymbol{v}_2)$. This determination is sufficient to identify the causal relationships between observable variables $\boldsymbol{x}$ as described by Equation (21). Consequently, one can safely intervene in the observable variables of the ODE system and make reliable causal inferences, despite the fact that matrix $B$ cannot be identified under this definition.

**Theorem E.2.** *For $\boldsymbol{x}_0 \in \mathbb{R}^d, \boldsymbol{z}_0 \in \mathbb{R}^p, A \in \mathbb{R}^{d\times d}, B \in \mathbb{R}^{d\times p}$, and $\boldsymbol{v}_1, \boldsymbol{v}_2 \in \mathbb{R}^p$, the ODE system (2) is $\boldsymbol{\theta}$-identifiable if and only if assumption **E1** is satisfied.*

> **E1** *the set of vectors $\{\boldsymbol{y}_0, F\boldsymbol{y}_0, \ldots, F^{d+2}\boldsymbol{y}_0\}$ is linearly independent, where $\boldsymbol{y}_0 = [\boldsymbol{x}_0^T, 0, 1, 1]^T$, and*
>
> $$F = \begin{bmatrix} A & B\boldsymbol{v}_2 & -B\boldsymbol{v}_1 & B\boldsymbol{z}_0 + B\boldsymbol{v}_1 \\ \mathbf{0}_d & 0 & 1 & 0 \\ \mathbf{0}_d & -1 & 0 & 0 \\ \mathbf{0}_d & 0 & 0 & 0 \end{bmatrix},$$
>
> $\mathbf{0}_d$ *denotes a $d$ dimensional zero row vector.*

The proof of Theorem E.2 is presented below. Condition **E1** is both sufficient and necessary, indicating, from a geometric perspective, that the vector $\boldsymbol{y}_0$ is not contained in an $F$-invariant proper subspace of $\mathbb{R}^{d+3}$.

*Proof.* Set

$$\boldsymbol{y}(t) = \begin{bmatrix} \boldsymbol{x}(t) \\ sin(t) \\ cos(t) \\ 1 \end{bmatrix},$$

we see that $\boldsymbol{y}(t) \in \mathbb{R}^{d+3}$, and the first derivative of $\boldsymbol{y}(t)$ w.r.t. time $t$ can be expressed as

$$\dot{\boldsymbol{y}}(t) = \begin{bmatrix} \dot{\boldsymbol{x}}(t) \\ cos(t) \\ -sin(t) \\ 0 \end{bmatrix} = \underbrace{\begin{bmatrix} A & B\boldsymbol{v}_2 & -B\boldsymbol{v}_1 & B\boldsymbol{z}_0 + B\boldsymbol{v}_1 \\ \boldsymbol{0}_d & 0 & 1 & 0 \\ \boldsymbol{0}_d & -1 & 0 & 0 \\ \boldsymbol{0}_d & 0 & 0 & 0 \end{bmatrix}}_{F} \underbrace{\begin{bmatrix} \boldsymbol{x}(t) \\ sin(t) \\ cos(t) \\ 1 \end{bmatrix}}_{\boldsymbol{y}(t)},$$

where $\boldsymbol{0}_d$ denotes a $d$ dimensional zero row vector. Obviously,

$$\boldsymbol{y}(0) = [\boldsymbol{x}_0^T, 0, 1, 1]^T = \boldsymbol{y}_0 \,.$$

Therefore, $\boldsymbol{y}(t)$ follows a homogeneous linear ODE system that can be expressed as:

$$\begin{aligned} \dot{\boldsymbol{y}}(t) &= F\boldsymbol{y}(t) \,, \\ \boldsymbol{y}(0) &= \boldsymbol{y}_0 \,, \end{aligned} \tag{22}$$

where $F \in \mathbb{R}^{(d+3)\times(d+3)}$. Worth noting that all state variables in the ODE system (22) are observable. Then according to Lemma 2.1, the system (22) is $(\boldsymbol{y}_0, F)$-identifiable if and only if condition **E1** stated in Theorem E.2 is satisfied. That is, under assumption **E1**, the trajectory $\boldsymbol{y}(\cdot; \boldsymbol{y}_0, F)$ uniquely determines both $\boldsymbol{y}_0$ and matrix $F$. Consequently, it also uniquely determines $(\boldsymbol{x}_0, A, B\boldsymbol{z}_0, B\boldsymbol{v}_1, B\boldsymbol{v}_2)$, thus establishing that the ODE system (2) is $\boldsymbol{\theta}$-identifiable if and only if assumption **E1** is satisfied. $\square$

# F An alternative approach to identifying matrices $B$ and $G$ in the ODE system (3)

## F.1 Identifiability condition from *2p* controllable whole trajectories

Recall that $\boldsymbol{z}_0$ denotes the initial condition of the latent variables in the ODE system (3). We further specify the initial condition of the latent variable $z_j$ as $z_{0j}$ for $j = 1, \ldots, p$. Assume that it is possible to control the initial condition of each latent variable, $z_{0j}$, independently. Specifically, for each experiment, researchers can intervene in the initial condition of a latent variable, denoted as $z_{0j}^*$. The value of $z_{0j}^*$ is treated as a given value. Under this intervention, the initial conditions of the latent variables are adjusted to $[z_{01}, \ldots, z_{0j}^*, \ldots, z_{0p}]^T$, which we denote as $\tilde{\boldsymbol{z}}_{0j}$.

To identify matrices $B$ and $G$, it is necessary to have at least two intervened initial conditions for each latent variable, denoted as $z_{0j}^{*1}$ and $z_{0j}^{*2}$ for the latent variable $z_j$. Consequently, the corresponding intervened initial conditions for all latent variables can be represented as $\tilde{\boldsymbol{z}}_{0j}^1$ and $\tilde{\boldsymbol{z}}_{0j}^2$. Under these conditions, we present the definition of the identifiability of the ODE system (3).

**Definition F.1.** *Given $z_{0j}^{*1}, z_{0j}^{*2} \in \mathbb{R}$ for $j = 1, \ldots, p$, for $\boldsymbol{x}_0 \in \mathbb{R}^d, \boldsymbol{z}_0 \in \mathbb{R}^p, A \in \mathbb{R}^{d \times d}, B \in \mathbb{R}^{d \times p}$ and $G \in \mathbb{R}^{p \times p}$, under the latent DAG assumption, for all $\boldsymbol{x}_0' \in \mathbb{R}^d$, all $\boldsymbol{z}_0' \in \mathbb{R}^p$, all $A' \in \mathbb{R}^{d \times d}$, all $B' \in \mathbb{R}^{d \times p}$, and all $G' \in \mathbb{R}^{p \times p}$, we denote $\tilde{\boldsymbol{z}}_{0j}^i = [z_{01}, \ldots, z_{0j}^{*i}, \ldots, z_{0p}]^T$ and $(\tilde{\boldsymbol{z}}_{0j}')^i = [z_{01}', \ldots, z_{0j}^{*i}, \ldots, z_{0p}']^T$, we further denote $\boldsymbol{\eta}_j^i := (\boldsymbol{x}_0, \tilde{\boldsymbol{z}}_{0j}^i, A, B, G)$ and $(\boldsymbol{\eta}_j')^i := (\boldsymbol{x}_0', (\tilde{\boldsymbol{z}}_{0j}')^i, A', B', G')$ for $i = 1, 2$, we say the ODE system (3) is $\{\boldsymbol{\eta}_j^{1,2}\}_1^p$-identifiable: if $(\boldsymbol{x}_0, A, B, G) \neq (\boldsymbol{x}', A', B', G')$, it holds that $\exists i \in \{1, 2\}$ and $j \in \{1, \ldots, p\}$ such that $\boldsymbol{x}(\cdot; \boldsymbol{\eta}_j^i) \neq \boldsymbol{x}(\cdot; (\boldsymbol{\eta}_j')^i)$.*

Definition F.1 establishes the identifiability of the ODE system (3) from $2p$ whole trajectories $\boldsymbol{x}(\cdot; \boldsymbol{\eta}_j^i)$ with $i = 1, 2$ and $j = 1, \ldots, p$. According to this definition, both matrices $B$ and $G$ are identifiable. Based on this definition, we present the identifiability condition.

**Theorem F.1.** *Given $z_{0j}^{*1}, z_{0j}^{*2} \in \mathbb{R}$ with $z_{0j}^{*1} \neq z_{0j}^{*2}$ for $j = 1, \ldots, p$, for $\boldsymbol{x}_0 \in \mathbb{R}^d, \boldsymbol{z}_0 \in \mathbb{R}^p, A \in \mathbb{R}^{d \times d}, B \in \mathbb{R}^{d \times p}$ and $G \in \mathbb{R}^{p \times p}$, under the latent DAG assumption, the ODE system (3) is $\{\boldsymbol{\eta}_j^{1,2}\}_1^p$-identifiable if assumptions $\boldsymbol{B}_5$ and $\boldsymbol{B}_4$ are both satisfied.*

> **B5**: *each $\tilde{\boldsymbol{z}}_{0j}^i$ for $i = 1, 2$ and $j = 1, \ldots, p$, satisfies assumption **B1**. That is, if we set $\boldsymbol{\gamma}_j^i = A^p \boldsymbol{x}_0 + \sum_{k=0}^{p-1} A^{p-1-k} BG^k \tilde{\boldsymbol{z}}_{0j}^i$, then the set of vectors $\{\boldsymbol{\gamma}_j^i, A\boldsymbol{\gamma}_j^i, \ldots, A^{d-1}\boldsymbol{\gamma}_j^i\}$ is linearly independent for all $i = 1, 2$ and $j = 1, \ldots, p$.*

The proof of Theorem F.1 is presented below. Assumption **B5** ensures that the ODE system (3) is $\boldsymbol{\eta}_j^i$-identifiable for all $i = 1, 2$ and $j = 1, \ldots, p$. Consequently, $(\boldsymbol{x}_0, A, B\tilde{\boldsymbol{z}}_{0j}^i, BG\tilde{\boldsymbol{z}}_{0j}^i, \ldots, BG^{p-1}\tilde{\boldsymbol{z}}_{0j}^i)$ for all $i = 1, 2$ and $j = 1, \ldots, p$ is identifiable. Through straightforward calculations, the identifiability of matrix $B$ is established. To identify matrix $G$, assumption **B4** is required.

The assumption that the initial condition of each latent variable $z_i$ can be controlled independently is inspired by the "genetic single-node intervention" proposed in [32], where interventions can be made at each latent node individually. This assumption is relatively more relaxed compared to controlling the initial condition of all latent variables $\boldsymbol{z}$ simultaneously, as discussed in Subsection 4.3. However, this method requires $p$ more trajectories, totalling $2p$ trajectories, to identify matrices $B$ and $G$.

*Proof.* Under assumption **B5**, since each $\tilde{\boldsymbol{z}}_{0j}^i$ satisfies assumption **B1**. By Theorem 4.1, the ODE system (3) is $\boldsymbol{\eta}_j^i$-identifiable for all $i = 1, 2$ and $j = 1, \ldots, p$. Consequently,

$$(\boldsymbol{x}_0, A, B\tilde{\boldsymbol{z}}_{0j}^i, BG\tilde{\boldsymbol{z}}_{0j}^i, \ldots, BG^{p-1}\tilde{\boldsymbol{z}}_{0j}^i)$$

for all $i = 1, 2$ and $j = 1, \ldots, p$ is identifiable.

We express $B\tilde{\boldsymbol{z}}_{0j}^i$ as

$$B\tilde{\boldsymbol{z}}_{0j}^i = \begin{bmatrix} B_{11} & \dots & B_{1j} & \dots & B_{1p} \\ \vdots & \ddots & \vdots & \ddots & \vdots \\ B_{d1} & \dots & B_{dj} & \dots & B_{dp} \end{bmatrix} \begin{bmatrix} z_{01} \\ \vdots \\ z_{0j}^{*i} \\ \vdots \\ z_{0p} \end{bmatrix}.$$

We know that $B\tilde{\boldsymbol{z}}_{0j}^i \in \mathbb{R}^d$ is identifiable for $i = 1, 2$. Thus, the first entry of $B\tilde{\boldsymbol{z}}_{0j}^i$, denoted as $(B\tilde{\boldsymbol{z}}_{0j}^i)_1$, is identifiable and can be expressed as

$$(B\tilde{\boldsymbol{z}}_{0j}^1)_1 = B_{11}z_{01} + \dots + B_{1j}z_{0j}^{*1} + \dots + B_{1p}z_{0p}$$
$$(B\tilde{\boldsymbol{z}}_{0j}^2)_1 = B_{11}z_{01} + \dots + B_{1j}z_{0j}^{*2} + \dots + B_{1p}z_{0p}.$$

Since $z_{0j}^{*1}$ and $z_{0j}^{*2}$ are given values, we can easily calculate the value of $B_{1j}$. Similarly, one can calculate the values of $B_{mj}$ for all $m = 1, \dots, d$ and $j = 1, \dots, p$, thereby establishing the identifiability of matrix $B$.

In a similar manner, matrices $BG, BG^2, \dots, BG^{p-1}$ are also identifiable. Then, according to the proof D.4 of Theorem 4.3, the matrix $G$ is identifiable under assumption **B4**. $\qquad\square$

### F.2 Identifiability condition from discrete observations sampled from *2p* controllable trajectories

We further extend the identifiability analysis of the ODE system (3) to cases where only discrete observations from $2p$ controllable trajectories are available.

**Definition F.2.** *Given $z_{0j}^{*1}, z_{0j}^{*2} \in \mathbb{R}$ for $j = 1, \dots, p$, for $\boldsymbol{x}_0 \in \mathbb{R}^d, \boldsymbol{z}_0 \in \mathbb{R}^p, A \in \mathbb{R}^{d \times d}, B \in \mathbb{R}^{d \times p}$ and $G \in \mathbb{R}^{p \times p}$. For any $n \geqslant 1$, let $t_k, k = 1, \dots, n$ be any $n$ time points and $\boldsymbol{x}_{jk}^i := \boldsymbol{x}(t_k; \boldsymbol{\eta}_j^i)$ be the error-free observation of the trajectory $\boldsymbol{x}(\cdot; \boldsymbol{\eta}_j^i)$ at time $t_k$. Under the latent DAG assumption, we say the ODE system (3) is $\{\boldsymbol{\eta}_j^{1,2}\}_1^p$-identifiable from $\boldsymbol{x}_{j1}^i, \dots, \boldsymbol{x}_{jn}^i$, $i = 1, 2$ and $j = 1, \dots, p$, if for all $\boldsymbol{x}_0' \in \mathbb{R}^d$, all $\boldsymbol{z}_0' \in \mathbb{R}^p$, all $A' \in \mathbb{R}^{d \times d}$, all $B' \in \mathbb{R}^{d \times p}$, and all $G' \in \mathbb{R}^{p \times p}$ with $(\boldsymbol{x}_0, A, B, G) \neq (\boldsymbol{x}_0', A', B', G')$, it holds that $\exists i \in \{1, 2\}, j \in \{1, \dots, p\}$ and $k \in \{1, \dots, n\}$ such that $\boldsymbol{x}(t_k; \boldsymbol{\eta}_j^i) \neq \boldsymbol{x}(t_k; (\boldsymbol{\eta}_j')^i)$.*

Based on Definition F.2 we present the identifiability condition.

**Theorem F.2.** *Given $z_{0j}^{*1}, z_{0j}^{*2} \in \mathbb{R}$ with $z_{0j}^{*1} \neq z_{0j}^{*2}$ for $j = 1, \dots, p$, for $\boldsymbol{x}_0 \in \mathbb{R}^d, \boldsymbol{z}_0 \in \mathbb{R}^p, A \in \mathbb{R}^{d \times d}, B \in \mathbb{R}^{d \times p}$ and $G \in \mathbb{R}^{p \times p}$. We define new observation $\boldsymbol{y}_{jk}^i := [(\boldsymbol{x}_{jk}^i)^T, 1, t_k, t_k^2, \dots, t_k^{p-1}]^T \in \mathbb{R}^{d+p}$, for $i = 1, 2, j = 1, \dots, p$ and $k = 1, \dots, n$. Under the latent DAG assumption, the ODE system (3) is $\{\boldsymbol{\eta}_j^{1,2}\}_1^p$-identifiable from discrete observations $\boldsymbol{x}_{j1}^i, \dots, \boldsymbol{x}_{jn}^i$, $i = 1, 2$ and $j = 1, \dots, p$, if assumptions **C3** and **B4** are both satisfied.*

> **C3**: *for each $i \in \{1, 2\}, j \in \{1, \dots, p\}$ there exists $(d + p)$ $\boldsymbol{y}_{jk}^i$'s with indexes denoting as $\{k_{j1}^i, k_{j2}^i, \dots, k_{j,d+p}^i\} \subseteq \{1, 2, \dots, n\}$, such that the set of vectors $\{\boldsymbol{y}_{jk_{j1}^i}^i, \boldsymbol{y}_{jk_{j2}^i}^i, \dots, \boldsymbol{y}_{jk_{j,d+p}^i}^i\}$ is linearly independent.*

The proof of Theorem F.2 is presented below. Assumption **C3** ensures that the ODE system (3) is $\boldsymbol{\eta}_j^i$-identifiable from discrete observations $\boldsymbol{x}_{j1}^i, \dots, \boldsymbol{x}_{jn}^i$ for all $i = 1, 2$ and $j = 1, \dots, p$. As in Subsection F.1, matrix $B$ is identifiable. Then, under assumption **B4**, matrix $G$ is also identifiable.

*Proof.* Under assumption **C3**, for each $i \in \{1, 2\}$ and $j \in \{1, \dots, p\}$, the corresponding observations satisfy assumption **C1**. Based on Theorem 4.2, the ODE system (3) is $\boldsymbol{\eta}_j^i$-identifiable for all $i = 1, 2$ and $j = 1, \dots, p$. Consequently,

$$(\boldsymbol{x}_0, A, B\tilde{\boldsymbol{z}}_{0j}^i, BG\tilde{\boldsymbol{z}}_{0j}^i, \dots, BG^{p-1}\tilde{\boldsymbol{z}}_{0j}^i)$$

for all $i = 1, 2$ and $j = 1, \dots, p$ is identifiable.

Following the proof of Theorem F.1, matrix $B$ is identifiable. Under assumption **B4**, matrix $G$ is also identifiable. $\qquad\square$

# G More simulation results

In this section, we present additional simulation results for higher-dimensional cases, along with simulations that incorporate a variety of ground-truth parameter configurations.

## G.1 Higher dimensional cases

In this subsection, for the $\boldsymbol{\eta}$-(un)identifiable cases of the ODE system (3), we provide a case with $d = 5$ and $p = 5$. The true underlying parameters of the systems are provided below. Initial parameter values are set to the true parameters plus a random value drawn from a uniform distribution $U(-0.14, 0.14)$ for each replication. To ensure reliability in the estimation results, we perform 50 independent random replications for each configuration, reporting the mean and variance of the squared error in Table 5.

$$
A = \begin{bmatrix} 2 & -2 & 1 & 1 & 1 \\ -1 & 1 & 0 & 2 & -2 \\ -2 & 2 & 0 & -1 & -2 \\ -1 & -1 & -2 & -1 & 2 \\ 1 & -2 & 1 & -2 & 0 \end{bmatrix}, \quad B = \begin{bmatrix} 1 & -2 & -1 & 1 & 1 \\ 1 & -2 & -1 & -1 & -1 \\ -2 & 0 & 2 & 1 & 1 \\ 0 & 2 & 0 & -2 & -2 \\ 2 & -2 & 2 & -1 & 2 \end{bmatrix},
$$

$$
G = \begin{bmatrix} 0 & 0 & 0 & -2 & -1 \\ 0 & 0 & -1 & 1 & 1 \\ 0 & 0 & 0 & 1 & 2 \\ 0 & 0 & 0 & 0 & 2 \\ 0 & 0 & 0 & 0 & 0 \end{bmatrix}, \quad A' = \boldsymbol{I}_5, \quad \boldsymbol{x}_0 = \begin{bmatrix} 2 \\ -2 \\ 2 \\ 1 \\ 0 \end{bmatrix}, \quad \boldsymbol{z}_0 = \begin{bmatrix} -2 \\ -1 \\ -1 \\ 1 \\ -2 \end{bmatrix},
$$

$\boldsymbol{\eta}$-identifiable: $\boldsymbol{\eta} = (\boldsymbol{x}_0, \boldsymbol{z}_0, A, B, G)$, unidentifiable: $\boldsymbol{\eta} = (\boldsymbol{x}_0, \boldsymbol{z}_0, A', B, G)$.

$\boldsymbol{I}_j$ denotes a $j \times j$ identity matrix.

Table 5: MSEs of the $\boldsymbol{\eta}$-(un)identifiable cases of the ODE (3) with $d = 5, p = 5$

| | $n$ | $A$ | $B\boldsymbol{z}_0$ | $BG\boldsymbol{z}_0$ | $BG^2\boldsymbol{z}_0$ | $BG^3\boldsymbol{z}_0$ | $BG^4\boldsymbol{z}_0$ |
|---|---|---|---|---|---|---|---|
| **Identifiable** | 10 | 0.0148 (±0.0006) | 0.3911 (±0.5989) | 0.9624 (±3.9249) | 0.7316 (±1.8971) | 0.1037 (±0.0374) | 0.0096 (±0.0003) |
| | 100 | 0.0059 (±4.01E-05) | 0.1529 (±0.0277) | 0.1726 (±0.0541) | 0.2447 (±0.0748) | 0.0212 (±0.0007) | 0.0012 (±1.10E-05) |
| | 1000 | 0.0053 (±2.92E-05) | 0.1394 (±0.0200) | 0.1241 (±0.0251) | 0.2119 (±0.0479) | 0.0164 (±0.0004) | 0.0004 (±6.00E-07) |
| **Unidentifiable** | 10 | 0.0853 (±0.0075) | 1.0067 (±1.3518) | 3.7422 (±55.8402) | 2.7696 (±24.5043) | 0.9229 (±2.7959) | 0.0508 (±0.0111) |
| | 100 | 0.0357 (±0.0019) | 0.4091 (±0.3812) | 1.0428 (±2.1792) | 0.9782 (±5.3654) | 0.3871 (±0.6747) | 0.0256 (±0.0032) |
| | 1000 | 0.0332 (±0.0017) | 0.3286 (±0.1824) | 0.7123 (±1.8836) | 0.9782 (±2.3163) | 0.5487 (±0.9240) | 0.0393 (±0.0047) |

For $\{\boldsymbol{\eta}_i\}_1^p$-(un)identifiable cases of the ODE system (3), we consider a case with $d = 10$ and $p = 5$. To accelerate estimation, sparsity is introduced in the parameter matrices by randomly setting 70, 35, and 20 entries in matrices $A$, $B$ and $G$, respectively, as zero. The true underlying parameters of the systems are provided below. Initial parameter values are set to the true parameters plus a random value drawn from a uniform distribution $U(-0.1, 0.1)$ for each replication. To ensure reliability in the estimation results, we perform 50 independent random replications for each configuration,

reporting the mean and variance of the squared error in Table 6.

$$A = \begin{bmatrix} 0 & 0 & -2 & -1 & 1 & 2 & 0 & -2 & -1 & 0 \\ 0 & 0 & 0 & 0 & 0 & 2 & 0 & -2 & 2 & 0 \\ 0 & 0 & 0 & 0 & 0 & 2 & 0 & 1 & 1 & 0 \\ 0 & 0 & 0 & -1 & 0 & 0 & 1 & 0 & -2 & 0 \\ 2 & 0 & 0 & -1 & 0 & -2 & 0 & 0 & -1 & 1 \\ 2 & 0 & 0 & 0 & 0 & 0 & 0 & 2 & 0 & -2 \\ 0 & 2 & 0 & 0 & 0 & 0 & 0 & 0 & 0 & 0 \\ -2 & -1 & 0 & 0 & 0 & 0 & 0 & 0 & 0 & 0 \\ 0 & 0 & 0 & -2 & 0 & 0 & 0 & 0 & 0 & -2 \\ 0 & 0 & 0 & 0 & -1 & 0 & 0 & 0 & 0 & -1 \end{bmatrix}, \quad B = \begin{bmatrix} -1 & 0 & 0 & 0 & 2 \\ 0 & -1 & 0 & 2 & 0 \\ 0 & -1 & 0 & 0 & 0 \\ 0 & 0 & 0 & 1 & 1 \\ 0 & 0 & 0 & 0 & 0 \\ 0 & 1 & 0 & 0 & 1 \\ 0 & 0 & -1 & 0 & 0 \\ 1 & 0 & 0 & 0 & 0 \\ 1 & 0 & 0 & 0 & -1 \\ -1 & 0 & 0 & 0 & -1 \end{bmatrix},$$

$$G = \begin{bmatrix} 0 & 1 & -1 & 0 & 2 \\ 0 & 0 & 2 & 0 & 0 \\ 0 & 0 & 0 & -1 & 0 \\ 0 & 0 & 0 & 0 & 0 \\ 0 & 0 & 0 & 0 & 0 \end{bmatrix}, \quad A' = \boldsymbol{I}_{10},$$

$\boldsymbol{x}_0 = \begin{bmatrix} -2 & 0 & 0 & -2 & 2 & -1 & 1 & 0 & 1 & 1 \end{bmatrix}^\top, \quad \boldsymbol{z}_0^{*i} = \boldsymbol{e}_i, \text{for } i = 1, \ldots, 5.$

$\{\boldsymbol{\eta}_i\}_1^p$-identifiable: $\boldsymbol{\eta}_i = (\boldsymbol{x}_0, \boldsymbol{z}_0^{*i}, A, B, G)$, unidentifiable: $\boldsymbol{\eta}_i = (\boldsymbol{x}_0, \boldsymbol{z}_0^{*i}, A', B, G)$.

$\boldsymbol{e}_i$ stands for a $p$-dimensional vector, with the $i$-th entry being 1 and the other entries being 0.

Table 6: MSEs of the $\{\boldsymbol{\eta}_i\}_1^p$-(un)identifiable cases of the ODE (3) with $d = 10, p = 5$

| $n$ | Identifiable | | | Unidentifiable | | |
|---|---|---|---|---|---|---|
| | $A$ | $B$ | $G$ | $A$ | $B$ | $G$ |
| 10 | 1.53E-11 (±2.36E-21) | 2.49E-10 (±6.30E-19) | 3.01E-10 (±9.20E-19) | 0.8345 (±0.6268) | 0.2118 (±0.0260) | 0.0037 (±0.0002) |
| 30 | 9.15E-13 (±4.45E-24) | 1.49E-11 (±1.18E-21) | 1.80E-11 (±1.73E-21) | 0.7216 (±0.4099) | 0.1952 (±0.0156) | 1.25E-21 (±5.18E-41) |
| 50 | 9.64E-14 (±1.29E-25) | 1.57E-12 (±3.43E-23) | 1.90E-12 (±5.02E-23) | 0.6510 (±0.2251) | 0.2211 (±0.0278) | 0.0042 (±0.0003) |

Tables 5 and 6 present results similar to those in Tables 1 and 2, providing strong empirical support for the validity of our proposed identifiability conditions.

## G.2 Various true parameters

To further support our proposed identifiability conditions, we conduct additional simulations incorporating a variety of ground-truth parameter configurations, rather than a fixed underlying parameter set. Specifically, for each simulation run, a unique ground-truth parameter configuration was generated using different random seeds, and we subsequently reported the mean and variance of the squared error across all results. For the low-dimensional $\boldsymbol{\eta}$ and $\{\boldsymbol{\eta}_i\}_1^p$ (un)identifiable cases, we perform 100 replications, while for the higher-dimensional cases, we perform 50 replications. Additionally, in the $\{\boldsymbol{\eta}_i\}_1^p$-(un)identifiable cases, we initialize the parameter values as the true parameters plus a random value drawn from $U(-0.1, 0.1)$ for the $d = 3, p = 3$ case and from $U(-0.05, 0.05)$ for the $d = 10, p = 5$ cases. For the $\boldsymbol{\eta}$-(un)identifiable cases, the initialization settings are the same as those used in the fixed-parameter configurations.

The simulation results are presented in Tables 7, 8, 9, and 10. Across all these tables, parameter estimates in the identifiable cases are notably more accurate than in the unidentifiable cases, providing strong empirical support for the validity of our proposed identifiability conditions.

It is noteworthy, however, that even in theoretically identifiable cases, certain scenarios emerge where parameter identification is challenging in practice; we refer to these as hard estimate cases. In these instances, estimates may deviate significantly from satisfactory values, similar to challenges encountered in fully observable ODE systems (1) as discussed in [28]. Consequently, for identifiable cases

with varying true parameter configurations, the results are less precise than those for corresponding fixed-parameter cases, due to the inclusion of some hard estimate instances. Investigating the practical identifiability of the ODE system (3) remains an intriguing direction for future research.

Table 7: MSEs of the $\eta$-(un)identifiable cases of the ODE (3) - with various true parameters

| $n$ | Identifiable | | | | Unidentifiable | | | |
|---|---|---|---|---|---|---|---|---|
| | $A$ | $Bz_0$ | $BGz_0$ | $BG^2z_0$ | $A$ | $Bz_0$ | $BGz_0$ | $BG^2z_0$ |
| 10 | 0.0060 ($\pm$0.0008) | 0.0157 ($\pm$0.0036) | 0.1698 ($\pm$0.5665) | 0.2297 ($\pm$1.1053) | 0.0691 ($\pm$0.0203) | 0.2720 ($\pm$0.5914) | 1.3133 ($\pm$7.4471) | 0.6622 ($\pm$8.5348) |
| 100 | 0.0026 ($\pm$9.27E-05) | 0.0108 ($\pm$0.0022) | 0.0820 ($\pm$0.1159) | 0.1287 ($\pm$0.7042) | 0.0283 ($\pm$0.0031) | 0.1003 ($\pm$0.0441) | 0.4880 ($\pm$2.6547) | 0.2649 ($\pm$1.6631) |
| 500 | 0.0020 ($\pm$6.48E-05) | 0.0092 ($\pm$0.0023) | 0.0870 ($\pm$0.1941) | 0.0705 ($\pm$0.1179) | 0.0227 ($\pm$0.0018) | 0.1061 ($\pm$0.0672) | 0.5015 ($\pm$3.0811) | 0.2574 ($\pm$2.0779) |

Table 8: MSEs of the $\{\eta_i\}_1^p$-(un)identifiable cases of the ODE (3) - with various true parameters

| $n$ | Identifiable | | | Unidentifiable | | |
|---|---|---|---|---|---|---|
| | $A$ | $B$ | $G$ | $A$ | $B$ | $G$ |
| 10 | 0.0006 ($\pm$2.21E-5) | 1.89E-5 ($\pm$3.55E-8) | 0.0009 ($\pm$6.71E-5) | 0.0861 ($\pm$0.1773) | 0.0088 ($\pm$0.0020) | 0.0101 ($\pm$0.0045) |
| 30 | 0.0006 ($\pm$2.20E-5) | 1.87E-5 ($\pm$3.47E-8) | 0.0010 ($\pm$6.64E-5) | 0.0789 ($\pm$0.1280) | 0.0092 ($\pm$0.0028) | 0.0104 ($\pm$0.0046) |
| 50 | 0.0006 ($\pm$2.21E-5) | 1.88E-5 ($\pm$3.51E-8) | 0.0009 ($\pm$6.67E-5) | 0.0503 ($\pm$0.0430) | 0.0063 ($\pm$0.0006) | 0.0114 ($\pm$0.0047) |

Table 9: MSEs of the $\eta$-(un)identifiable cases of the ODE (3) with $d = 5, p = 5$ - with various true parameters

| | $n$ | $A$ | $Bz_0$ | $BGz_0$ | $BG^2z_0$ | $BG^3z_0$ | $BG^4z_0$ |
|---|---|---|---|---|---|---|---|
| Identifiable | 10 | 0.0144 ($\pm$0.0004) | 0.1215 ($\pm$0.0757) | 1.4643 ($\pm$8.3976) | 2.1890 ($\pm$54.9706) | 1.8254 ($\pm$48.7033) | 0.4826 ($\pm$5.7127) |
| | 100 | 0.0041 ($\pm$4.55E-05) | 0.0395 ($\pm$0.0092) | 0.2850 ($\pm$0.1739) | 0.3891 ($\pm$0.4936) | 0.2078 ($\pm$0.2950) | 0.0239 ($\pm$0.0024) |
| | 1000 | 0.0032 ($\pm$3.26E-05) | 0.0337 ($\pm$0.0049) | 0.1934 ($\pm$0.0686) | 0.2242 ($\pm$0.2180) | 0.1197 ($\pm$0.0712) | 0.0181 ($\pm$0.0014) |
| Unidentifiable | 10 | 0.0740 ($\pm$0.0047) | 0.4599 ($\pm$0.4841) | 2.8628 ($\pm$9.5476) | 1.8743 ($\pm$8.6653) | 0.4834 ($\pm$1.2606) | 0.0334 ($\pm$0.0147) |
| | 100 | 0.0263 ($\pm$0.0031) | 0.2142 ($\pm$0.1869) | 1.1678 ($\pm$8.2277) | 1.2354 ($\pm$9.4970) | 0.2878 ($\pm$0.8655) | 0.0193 ($\pm$0.0052) |
| | 1000 | 0.0142 ($\pm$0.0003) | 0.1389 ($\pm$0.0463) | 0.6979 ($\pm$1.2080) | 0.6701 ($\pm$1.5228) | 0.0732 ($\pm$0.0336) | 0.0062 ($\pm$0.0003) |

Table 10: MSEs of the $\{\boldsymbol{\eta}_i\}_1^p$-(un)identifiable cases of the ODE (3) with $d = 10, p = 5$ - with various true parameters

| $n$ | Identifiable | | | Unidentifiable | | |
|---|---|---|---|---|---|---|
| | $A$ | $B$ | $G$ | $A$ | $B$ | $G$ |
| 10 | 0.0044 ($\pm$0.0001) | 0.0350 ($\pm$0.0098) | 0.0287 ($\pm$0.0053) | 0.6266 ($\pm$0.1524) | 0.1310 ($\pm$0.0269) | 0.0054 ($\pm$0.0004) |
| 30 | 0.0067 ($\pm$0.0005) | 0.1258 ($\pm$0.5097) | 0.0315 ($\pm$0.0104) | 0.5833 ($\pm$0.2085) | 0.1058 ($\pm$0.0114) | 0.0021 ($\pm$8.79E-05) |
| 50 | 0.0033 ($\pm$5.66E-05) | 0.0323 ($\pm$0.0103) | 0.0354 ($\pm$0.0084) | 0.5193 ($\pm$0.0982) | 0.1108 ($\pm$0.0146) | 0.0021 ($\pm$9.02E-05) |

