# OpenReview forum: "Identifiability Analysis of Linear ODE Systems with Hidden Confounders"
_NeurIPS.cc/2024/Conference — NeurIPS 2024 poster_

### Official Review · Reviewer_LnTR · 2024-07-11

**Soundness:** 3
**Presentation:** 2
**Contribution:** 3
**Rating:** 6
**Confidence:** 2

**Summary:**

This paper provides the identifiability analysis of linear Ordinary Differential Equation (ODE) systems, particularly in scenarios where latent variables interact with the system. In detail, it investigates two specific cases. In the first scenario, latent confounders do not exhibit causal relationships, but their evolution follows specific functional forms, such as polynomial functions of time. The analysis is then extended to a second, more complex scenario, where hidden confounders have causal dependencies described by a Directed Acyclic Graph (DAG). The authors perform a series of simulations to substantiate their theoretical results.

**Strengths:**

This paper makes a significant contribution by extending the understanding of identifiability in linear ODE systems to include cases with latent variables, thereby enhancing the reliability of causal inferences in more complex systems. The simulated experimental results provide strong support for the theoretical findings, making this a robust and valuable study in the field.

**Weaknesses:**

1. The symbols $x'$ and $A'$ in the paper are not clearly defined. This may lead to confusion for readers when understanding the derivation process and the results. It is recommended to clearly define and explain these symbols in the paper.
2. The assumptions and theorems lack intuitive explanations. It would be better to provide some intuitive explanations or examples after each assumption and theorem to help readers better understand the essence of these theories and their roles in practical applications.

**Questions:**

How practical are the proposed assumptions in the paper? Can the authors discuss their validity and design experiments to test their validity for real datasets?

**Limitations:**

See the weaknesses above.

---

> ### Author Rebuttal · Authors · 2024-08-07
>
> Thank you for your valuable comments. We have addressed each of your comments as follows. Additionally, we have revised our manuscript in accordance with your suggestions, and we believe that the quality has been significantly enhanced as a result of your insightful input.
>
> ## Answers to weaknesses:
>
> >**W1:** Explain the symbols $\boldsymbol{x}_0'$ and $A'$.
>
> **A:** Thank you for pointing this out. The symbols $\boldsymbol{x}'_0$ and $A'$ are first introduced in our paper in Definition 2.1. To enhance clarity, we have included an explanation in the subsequent paragraph, which reads as ``We use $\boldsymbol{x}_0'$ and $A'$ to distinguish other system parameters from the true system parameters $\boldsymbol{x}_0$ and $A$; $\boldsymbol{x}_0'$ and $A'$ can represent any $d$-dimensional initial conditions and any $d\times d$ parameter matrices, respectively."
>
> >**W2:** Provide intuitive explanations for the assumptions and theorems.
>
> **A:**  Thank you for your valuable suggestion. We have provided intuitive explanations for each assumption as they are introduced. The added explanations are as follows (_please refer to **Table 1** in the uploaded PDF file for the summary of the proposed conditions_):
>
> 1.  **For condition **A1** in Theorem 3.1 and condition **B1** in Theorem 4.1:** These conditions are the same as the one stated in Lemma 2.1 for the fully observable ODE system (1), but with a different initial condition $\boldsymbol{\beta}$ or $\boldsymbol{\gamma}$. We have added the intuitive explanation of this condition in the paragraph following Lemma 2.1, since it is where this condition was first introduced. The added explanation reads:
>
>     "From a geometric perspective, the set of vectors stated in Lemma 2.1 being linearly independent indicates that the initial condition $\boldsymbol{x}_0$ is not contained in an $A$-invariant proper subspace of $\mathbb{R}^d$. Intuitively, this means the trajectory of this system started from $\boldsymbol{x}_0$ spans the entire $d$-dimensional state space. That is, our observations cover information on all dimensions of the state space, thus rendering the identifiability of the system."
>
> 2. **For condition **C1** in Theorem 4.2:** The added explanation reads:
>
>     "Under the latent DAG assumption, we can transfer the ODE system (3), which includes hidden confounders, into a $(d+p)$-dimensional fully observable ODE system (1) through the augmented state $\boldsymbol{y}(t)$. Condition **C1** indicates that our observations span the entire $(d+p)$-dimensional state space, thus rendering the system identifiable."
>
> 3. **For conditions **B2, B3, B4** in Theorem 4.3:** Condition **B2** is the same as **B1**, for which the explanation has been added. Conditions **B3, B4** are straightforward and do not require additional intuitive explanations.
>
> 4. **For condition **C2** in Theorem 4.4:** This condition is the same as **C1**, for which the explanation has been added.
>
> In addition, we have included a table to summarize all the notations (please refer to our comment to reviewer Te42) and another table to outline the proposed identifiability conditions (available in the uploaded PDF file). We believe these additions significantly enhance the clarity and readability of our manuscript, aiding readers in better understanding the essence of our proposed theories.
>
> ## Answer to question:
>
> >**Q1:** Practical validity of the proposed theoretical results.
> >
> **A:** Thank you for your question. The primary assumption in our manuscript is the **latent DAG** assumption, which is standard in causality studies. For a detailed discussion on the practicality and reasonableness of this assumption, please refer to our response to W2 from Reviewer 1wF2.
>
> In addition to the **latent DAG** assumption, the other assumptions are quite mild:
> 1. **Conditions in Theorem 3.1, 4.1, 4.2:** These conditions are both sufficient and **necessary** and cannot be relaxed further in our linear ODE setup. As we have mentioned conditions **A1** and **B1** align with the condition stated in Lemma 2.1, which is the set of vectors $\\{\boldsymbol{x}_0, A\boldsymbol{x}_0, \ldots, A^{d-1}\boldsymbol{x}_0\\}$ being linearly independent (denote it as condition **A0**). This condition is **generic** as noted in [40], meaning that the set of system parameters that violate this condition has Lebesgue measure zero. Intuitively, condition **A0** is satisfied for **almost all** combinations of $\boldsymbol{x}_0$ and $A$. Once condition **B1** is satisfied, then one can always find observations on the same trajectory satisfying condition **C1**, rendering condition **C1** also mild.
> 2. **Conditions in Theorem 4.3 and 4.4:** These conditions are sufficient but not necessary. Identifying parameter matrices $B$ and $G$ requires additional observations (trajectories) and assumptions. Condition **B2** and **C2** mirror the mild conditions **B1** and **C1**, respectively. Condition **B3** is experimentally controllable and trivially satisfied, while condition **B4**, similar to **A0**, is **generic**. Thus, all proposed conditions are practical and not restrictive.
>
> Regarding real-world dataset experiments, we were unable to include them due to the unavailability of suitable data. However, we have added two real-world linear ODE examples (see the comment to Reviewer 1wF2) and additional higher-dimensional simulation cases to better support our theoretical results. The corresponding simulation results are presented in the uploaded PDF file. These simulations align with current studies on theoretical identifiability of linear ODE or SDE systems, which also **rely solely on simulations** (see [27, 33, 40, 41]).

---

> > ### Comment · Reviewer_LnTR · 2024-08-13
> >
> > Thanks for the responses. I will raise my score.

---

> > > ### Author Response · Authors · 2024-08-13
> > > **Response to Reviewer LnTR**
> > >
> > > Dear Reviewer LnTR,
> > >
> > > Thank you very much for raising your score. We greatly appreciate your recognition of our work and your valuable comments.
> > >
> > > Sincerely,
> > >
> > > The authors

---

### Official Review · Reviewer_Te42 · 2024-07-11

**Soundness:** 3
**Presentation:** 3
**Contribution:** 3
**Rating:** 6
**Confidence:** 4

**Summary:**

This paper studies the problem of identifiability analysis of linear Ordinary Differential Equation (ODE) systems. It focuses mainly on the scenarios where some variables in the system remain latent to the learner. This paper aims to address this challenge by studying identifiability analysis in two classes of linear ODE systems with latent confounders. The first case considers latent confounders that are mutually independent, and the second case includes correlated latent confounders. The authors conduct detailed identifiability analyses for both systems and propose sufficient identification conditions. Simulation results support the theoretical findings.

**Strengths:**

- The paper is well-written and clearly organized. The authors clearly stated all the necessary assumptions.
- Identifiability analysis is an important problem in causal inference and control theory. Most of the existing methods in causal inference literature focus on the acyclic systems without feedback loops. On the other hand, methods in control theory often assume there are no unobserved confounders in the system. This paper attempts to close this gap by studying causal identification in linear ODE systems with latent confounders. It could have a significant impact across disciplines, including AI, econometrics, and environmental science.

**Weaknesses:**

- This paper is dense, and it could be difficult for readers unfamiliar with ODE analysis. It could be recommended that the authors could include an additional table summarizing the notations. Also, a table summarizing the identification conditions would also be helpful.
- Simulations are performed on relatively simple synthetic instances. It would be interesting to see how the result scale to a more complex system.

**Questions:**

- How does the agent obtain the matrix $G$ from the latent DAG assumption? Could the author elaborate on this?

**Limitations:**

The authors have adequately addressed the limitations of the paper.

---

> ### Author Rebuttal · Authors · 2024-08-07
>
> Thank you for your valuable comments. We have addressed each of your comments as follows. Additionally, we have revised our manuscript in accordance with your suggestions, and we believe that the quality has been significantly enhanced as a result of your insightful input.
>
> ## Answers to weaknesses:
>
> >**W1:** Include tables that summarize the notations and identifiability conditions.
>
> **A:** Thank you for your valuable suggestion. In accordance with your recommendation, we have included a table to summarize all the notations (please refer to the following comment) and another table to summarize all the proposed identifiability conditions (available in the uploaded PDF file). We believe that these additions significantly enhance the clarity and readability of our manuscript.
>
> >**W2:** Scale the simulations to a more complex system.
>
> **A:** Thank you for your comment. We would like to clarify that, theoretically, our proposed identifiability conditions are applicable to any finite dimensions $d \geqslant 1$ and $p \geqslant 1$. In practical scenarios, as the system dimensions increase, the complexity of the system also escalates. Consequently, larger sample sizes and more advanced parameter estimation methods may be required to achieve satisfactory parameter estimates.
>
> In response to your suggestion, we have included additional simulation examples with increased complexity in our updated manuscript. For single trajectory identifiability validation (Theorems 4.1 and 4.2), we have added an example with $d=5, p=5$ (i.e., 5-dimensional observable variable and 5-dimensional latent variables, totalling 10 dimensions). For $p$ trajectory identifiability validation (Theorems 4.3 and 4.4), we have added an example with $d=10, p=5$ (i.e., 10-dimensional observable variables and 5-dimensional latent variables, totalling 15 dimensions). The results of these simulations are presented in the uploaded PDF file and provide empirical evidence supporting the validity of our proposed identifiability conditions in more complex systems.
>
> ## Answer to question:
>
> >**Q1:**  Elaborate on how the agent obtains matrix $G$.
>
> **A:** Thank you for your question. As discussed in our manuscript, the identifiability of matrix $G$ is established under the conditions stated in Theorem 4.3 and Theorem 4.4. The proof of Theorem 4.3, detailed in Appendix B.4, provides a comprehensive derivation of how matrix $G$ can be obtained. Due to the complexity and extensive use of notations, it is challenging to fully elaborate on this proof in a few plain sentences.
>
> To summarize, obtaining matrix $G$ requires not only the latent DAG assumption but also the assumptions B2, B3, and B4 outlined in Theorem 4.3. A crucial aspect of the latent DAG assumption is  that it allows the matrix $G\in \mathbb{R}^{p\times p}$ to be permuted into a strictly upper triangular form, such that $G^k = 0$ for all $k \geqslant p$, where $p$ is the dimension of the latent variables. Consequently, the states of hidden variables can be expressed as polynomial functions of time $t$, from which the identifiability conditions are derived.
>
> We hope this brief explanation provides some insight into the latent DAG assumption. For a detailed derivation of matrix $G$, we encourage you to refer to our proof in Appendix B.4.

---

> ### Author Response · Authors · 2024-08-07
> **Table for summarizing all the notations**
>
> Here, we provide the added table for summarizing all the notations.
>
> | Notation                              | Description                                                                                                                                |
> | ------------------------------------- | ------------------------------------------------------------------------------------------------------------------------------------------ |
> | $\boldsymbol{x/z}$   | observable/latent variables                 |
> | $x_i/z_i$ | the $i$-th observable/latent variable          |
> | $t$    | time            |
> | $t_j$   | the j-th time point    |
> | $\boldsymbol{x}(t)/\boldsymbol{z}(t)$ | state of observable/latent variable at time $t$                                                                                            |
> | $\boldsymbol{x}_j$                    | $\boldsymbol{x}(t_j)$, observable state at time $t_j$                                                                                      |
> | $\boldsymbol{x}_0/ \boldsymbol{z}_0$  | initial condition of observable/latent variable                                                                                            |
> | $\dot{\boldsymbol{x}}(t)$             | first derivative of $\boldsymbol{x}(t)$ w.r.t. time $t$                                                                                    |
> | $d$                                   | dimension of observable variables                                                                                                          |
> | $p$                                   | dimension of latent variables                                                                                                              |
> | $A,B,G$                               | constant parameter matrices defined in Eq.(2) and (3)                                                                                      |
> | $\boldsymbol{f}(t)$                   | Function of time $t$ defined in Eq.(2)                                                                                                     |
> | $\boldsymbol{v}_k$                    | constant parameter vector defined in Eq. (4)                                                                                               |
> | $\\{\boldsymbol{v}_k\\}_0^r$          | all the $\boldsymbol{v}_k$'s for $k=0,\ldots,r$                                                                                            |
> | $\boldsymbol{\theta}$                 | $:= (\boldsymbol{x}_0, \boldsymbol{z}_0, A, B, \\{\boldsymbol{v}_k\\}_0^r)$, the system parameter of ODE system (2)                        |
> | $\boldsymbol{\beta}$                  | a vector defined in Thm.3.1 A1                                                                                                             |
> | $\boldsymbol{y}(t)$                   | augmented state                                                                                                                            |
> | $\boldsymbol{y}_0$                    | initial condition of augmented variable                                                                                                    |
> | $\boldsymbol{\eta}$                   | $:= (\boldsymbol{x}_0, \boldsymbol{z}_0, A, B, G)$, the system parameter of ODE system (3)                                                 |
> | $\boldsymbol{\gamma}$                 | a vector defined in Thm.4.1 B1                                                                                                             |
> | $\boldsymbol{z}_0^{*}$                | given initial condition of latent variable                                                                                                 |
> | $\boldsymbol{z}_0^{*i}$               | the $i$-th given initial condition of latent variable                                                                                      |
> | $\boldsymbol{\eta}_i$                 | $:=(\boldsymbol{x}_0, \boldsymbol{z}_0^{*i}, A, B, G)$, the system parameter of ODE system (3)                                             |
> | $\boldsymbol{\gamma}_i$               | a vector defined in Thm 4.3 B2                                                                                                             |
> | $\boldsymbol{x}_{ij}$                 | $:= \boldsymbol{x}(t_j;\boldsymbol{\eta}_i)$, observable state of ODE system (3) with system parameter $\boldsymbol{\eta}_i$ at time $t_j$ |
> | $\boldsymbol{y}_{ij}$                 | augmented state of $\boldsymbol{x}_{ij}$ at time $t_j$                                                                                     |
> | $A', \boldsymbol{x}_0', \ldots$       | the alternative counterpart corresponding to $A,\boldsymbol{x}_0, \ldots$                                                                  |

---

> > ### Comment · Reviewer_Te42 · 2024-08-12
> > **Thank you for the response**
> >
> > I appreciate the authors' detailed response. While the simulations could still be improved, this paper proposes novel theoretical identification results in challenging problem settings, i.e.,  dynamic systems with hidden confounders and feedback. I will raise my confidence score.

---

> ### Author Response · Authors · 2024-08-12
> **Response to comment from Reviewer Te42**
>
> Dear Reviewer Te42,
>
> Thank you so much for raising your confidence score. Your recognition of our work means a great deal to us. In addition, regarding the simulations, we have added an additional simulation inspired by Reviewer 1wF2's comment. Specifically, we set different ground-truth parameter configurations by using different seeds. Due to time constraints, we have currently applied 10 different configurations to the single and multiple trajectory experiment with $d=3, p=3$. We will update these results to include 100 different configurations in our final manuscript, and we will also include higher-dimensional cases.
>
> The simulation results are provided in the comment to Reviewer 1wF2. Through these additional simulations, we have increased the diversity of our ground-truth examples, providing more convincing empirical support that our proposed theoretical results are suitable for any system parameter configurations that meet the proposed identifiability conditions. We believe that our simulation has been greatly improved by adding this set of simulations.
>
> Thank you again for increasing your confidence score and for your valuable comments.
>
>
> Sincerely,
>
> The Authors

---

### Official Review · Reviewer_1wF2 · 2024-07-12

**Soundness:** 3
**Presentation:** 3
**Contribution:** 2
**Rating:** 5
**Confidence:** 2

**Summary:**

The paper focuses on the (parameter) identifiability problem of linear ODE systems. The identifiability results of the existing work has been limited to fully-observable systems, i.e., with no latent variables. The paper analyzes the parameter identifiability of partially-observable linear ODEs with certain structure, that is $\dot{\mathbf{z}}(t) = \mathbf{0} \mathbf{x}(t) + G \mathbf{z}(t)$: (i) the observables don’t affect the time derivative of the latents, and (ii) the latent transition matrix $G$ is strictly upper-triangular (“DAG structure”). For this setup, they characterize the identifiability conditions for a single trajectory and multiple trajectories; in addition to the cases where these trajectories are observed in discrete-time steps. They evaluate the validity of the proposed identifiability conditions on a simulation study, by comparing the parameter estimation errors for identifiable and unidentifiable data generating processes.

**Strengths:**

* The paper extends the identifiability conditions of the previous work from fully-observable systems to partially-observable systems, which have practical value in real-world scenarios.
* The paper is very well-written and easy-to-follow despite being a theoretically heavy.

**Weaknesses:**

* The paper motivates the identifiability analysis with latent variables by its importance to practical scenarios in causal inference. Yet, the examples in the paper and its simulation setup are far away from being practical.
* In lines 142-143, the DAG assumption for the latent relationships is motivated as being common in causality studies. However, these studies only consider a static setup. From the provided references, it is not possible to see how feasible this assumption is for an ODE system.
* The simulation setup seems to be contrived, where the parameter means ($|x| \in {0,1,2}$) are chosen by hand with small uniform perturbations, $U(-0.1, 0.1)$ and $U(-0.3, 0.3)$. It is hard to say how much randomness this scenario creates. The simulation study would support the claim better if its setup shows more randomness.

**Questions:**

* To show the practical value of the paper, can you provide linear ODE examples having practical value in some fields, e.g., chemistry, etc? On these, the structural constraints could be assessed. Then, it could be checked if the identifiability conditions hold for the typical ranges of the real-world variables. In addition, these can be added to the simulation study where the identified parameters have real-world meaning, explaining certain intervention effects.
* Even though I understand what you mean by the "DAG structure", I think the graph considered here is not well defined. The variables do not affect each other as demonstrated in Figure 1, they affect each other's time derivatives. The nodes in the graphs in Figure 1 represent two things at the same time: the variable states and the time derivatives.
* What happens if you increase the system dimensionality in the simulation study? Currently, it is set to $d=3$.

**Limitations:**

* The main limitation seems to be the verification of the identifiability conditions in practical scenarios, as noted by the authors.

---

> ### Author Rebuttal · Authors · 2024-08-07
>
> Thank you for your valuable comments. We have addressed each of your comments as follows. Additionally, we have revised our manuscript in accordance with your suggestions, and we believe that the quality has been significantly enhanced as a result of your insightful input.
>
> ## Answers to weaknesses:
>
> >**W1:** Provide practical examples.
>
> **A:** Thank you for your comment. Since both this and your first question (Q1) address the practical value of our paper, we will respond to them together here.
>
> In response to your suggestion, we have included two real-world linear ODE examples. Due to the 6000-character limit, detailed descriptions of these two models are provided in the following comment, and the corresponding causal graphs are presented in the uploaded PDF file. As illustrated by the graphs, the structure of these models aligns well with our structural constraints.
>
> Regarding the applicability of the proposed identifiability condition and real-world dataset experiments, we kindly refer you to our detailed response to Q1 from Reviewer LnTR, which addresses this query comprehensively. Thank you for your understanding.
>
> >**W2:** Feasibility of the latent DAG assumption.
>
> **A:** Thank you for your comment. In response to your second question, we believe there may be some confusion regarding how the causal graph is defined within the context of an autonomous (time-invariant) ODE system. To address this, we will first clarify the causal graph in the context of an ODE system, which we hope will provide a clearer understanding of the feasibility of the latent DAG assumption.
>
> To enhance understanding, consider a general fully observable time-invariant ODE system  $\dot{\boldsymbol{x}}(t) = f(\boldsymbol{x}(t))$. In such systems, the derivatives of state variables $\dot{\boldsymbol{x}}(t)$ do not explicitly depend on time $t$. For an ODE system like this, we state there is a direct causal relationship from variable $x_j$ to variable $x_i$ if $\dot{x_i}$ is dependent on $x_j$, expressed as $\dot{x}_i(t)= f(x_j(t))$. As detailed in [24], the causal graph for such an ODE system is defined such that **each node represents a variable $x_i$, and there is a direct edge from $x_j$ to $x_i$ if and only if $\dot{x}_i$ depends on $x_j$**. This causal graph remains invariant over time in a time-invariant ODE system. Both ODE systems (2) and (3) in our manuscript are time-invariant, and the graphs in Figure 1 are well-defined within this context.
>
> We assert that the latent DAG assumption is reasonable for several reasons:
> 1.  Consider a particle of mass $m$ in a uniform gravitational field where the gravitational field exerts a constant force $F$ on the particle. The evolution of the particle's velocity (denoted as $v$) and position (denoted as $r$) can be described by a linear time-invariant ODE system:
>
>     $$ \dot{v}(t) = F/m, \dot{r}(t) = v. $$
>
>     The corresponding causal graph of this ODE system is $ v \rightarrow r$, which is a DAG. Hence, a DAG is a reasonable causal structure to describe ODE systems.
>
> 2. Since we focus on time-invariant ODE system analysis, the causal graph remains invariant with respect to time. Therefore, treating the causal graph as static and making a DAG assumption is a natural extension of traditional static causal studies.
>
> 3. Deriving identifiability conditions for causal models is a challenging problem. This is why the DAG assumption is adopted in static setups, even in fully observable cases. Similarly, deriving the identifiability conditions for linear ODE systems with hidden variables is difficult, and this field of study is still in its early stages. To our knowledge, our work is the first to systematically derive identifiability conditions for such ODE systems. Referring to well-established assumptions in traditional causal studies is a prudent starting point. Additionally, we allow for cycles and self-loops among observable variables and only assume that latent variables follow a DAG structure, which is relatively less restrictive compared to the classic DAG assumption in static setups. Without the latent DAG assumption, it is currently not feasible to derive identifiability conditions for linear ODE systems with hidden confounders.
>
> >**W3:** Regarding randomness of the simulation study.
>
> **A:** Thank you for your comment. Upon reviewing our uploaded codes in the supplementary material, you will see that all parameters in our simulation study are entirely randomly generated. Specifically, we used `randint(-2,3)` to generate the parameters for simplicity.
>
> We chose to set the initial parameter values close to the true parameter values with uniform perturbations $U(-0.1,0.1)$ and $U(-0.3,0.3)$ because the Nonlinear Least Squares (NLE) loss function associated with our simulation is non-convex. Initializing the parameters close to the true values helps the NLE converge to the true global minimum. Introducing more randomness into the initial parameter values would necessitate using a computationally intensive and time-consuming global optimization technique or another parameter estimation method.
>
> The primary objective of our simulation is to validate the proposed theoretical results rather than check or develop parameter estimation methods or techniques for ODEs. We believe that the current simulation results robustly support our theoretical claims. However, to address your concern and further substantiate our findings, we have included two additional higher-dimensional cases.
>
> ## Answers to questions:
>
> >**Q1:** Provide practical examples.
>
> **A:** Please refer to our response to W1.
>
> >**Q2:** Explain causal graph for an ODE system.
>
> **A:** Please refer to our response to W2.
>
> >**Q3:** Increase dimension in simulations.
>
> **A:** Thank you for your question. Due to the 6000-character limit, we kindly refer you to our response to W2 from Reviewer Te42, as it addresses the same query in detail. We appreciate your understanding.

---

> ### Author Response · Authors · 2024-08-07
> **Real-world linear ODE examples**
>
> ## Example 1: damped harmonic oscillators model
>
> Consider a one dimensional system of $D$ point masses $m_i (i = 1, \ldots, D)$ with positions $Q_i(t) \in \mathbb{R}$ and momenta $P_i(t) \in \mathbb{R}$. These masses are coupled by springs characterized by spring constants $k_i$ and equilibrium lengths $l_i$, and are subject to friction with a coefficient $b_i$, all while the end positions are fixed.
>
> The dynamics of this system are described by the following linear ODE system [24]:
> \begin{equation}
> \begin{split}
> \dot{P}_ i(t) &=k_i(Q_{i+1}(t)- Q_i(t) -l_i)-k_{i-1}(Q_i(t) -Q_{i-1}(t)-l_{i-1}) - b_i P_i(t)/m_i \\\\
> \dot{Q}_i(t) &= P_i(t)/m_i
> \end{split}
> \end{equation}
>
> where $Q_0(t) = 0$ and $Q_{D+1}(t) = L$ represent the fixed boundary conditions. External forces $F_j(t)$ (e.g., wind force or a varying magnetic field) can influence the entire system of coupled harmonic oscillators. These external forces can be modelled as latent variables with a constant derivative. Consequently, the system can be expressed as:
>
> \begin{equation}
> \begin{split}
> \dot{P}_ i(t) &= k_i(Q_{i+1}(t) - Q_i(t) -l_i)-k_{i-1}(Q_i(t) -Q_{i-1}(t)-l_{i-1}) - b_i P_i(t)/m_i + \sum_{j}\alpha_{ij} F_j(t)\\\\
> \dot{Q}_i(t) &= P_i(t)/m_i\\\\
> \dot{F}_j(t) &= c_j
> \end{split}
> \end{equation}
>
> where $\alpha_{ij}$ is a constant determining the effect of the external force $F_j(t)$ on the $i$-th mass, and $c_j$ is the constant rate of change of the external force $F_j(t)$. This model aligns well with our ODE system (2). An example causal graph illustrating this model is provided in the uploaded PDF file.
>
> ## Example 2: population model
>
> The growth of a population $P$ can be described by a linear ODE:
> \begin{equation*}
>     \dot{P}(t) = a P(t),
> \end{equation*}
> where $a$ is a constant representing the growth rate of the population. The system can also be influenced by latent variables $L_i$, such as environmental factors and food supply. Incorporating these latent influences, the system can be modelled as:
> \begin{equation*}
> \begin{split}
>     \dot{P}(t) &= a P(t) + b L_1(t) + c L_2 (t)\\\\
>     \dot{L}_1(t) &= l L_2(t)\\\\
>     \dot{L}_2(t) &= m
> \end{split}
> \end{equation*}
> where $a, b, c, l$ and $m$ are constants. Here, $L_1(t)$ represents the food supply, influenced by the environmental factor $L_2(t)$. $L_2(t)$ corresponds to an environmental factor, such as temperature or pollution levels, which changes steadily over time. This model aligns well with our ODE system (3).

---

> > ### Comment · Reviewer_1wF2 · 2024-08-09
> >
> > Dear authors,
> >
> > Thank you for your detailed response and your efforts to address my concerns. Your response is well-written and well-organized as your paper. However, I am still not convinced about my two main concerns.
> >
> > **1) Contrived simulation setup.**
> > * From (i) lines 275-276 “The true underlying parameters of the systems are provided below”, and (ii) the equations between lines 278-279, it seems to me that **the simulations have only a single ground-truth parameter configuration $\eta_{sim} = (\mathbf{x}_0, \mathbf{z}_0 A, B, G)$ provided in Eqs bwn lines 278-279.** Can you please clarify whether (i) you sample a single ground-truth parameter configuration ($\eta_{sim}$) or (ii) multiple configurations ($( \eta_{sim}^{(k)} )_{k=1}^K$) with $K$ different seeds?
> > * I still think that setting the initial values close to the ground-truth values with perturbations of U(-0.1, 0.1) and U(-0.3, 0.3) may not be sufficient to create enough randomness. Since the paper motivates itself by its importance to practical scenarios in causal inference and we cannot know the ground-truth values in practice beforehand, the paper should at least show what happens when we initialize the values randomly (or further away).
> > * I appreciate the results for higher dimensions $d=5, p=5$ and $d=10,p=5$. What is the motivation behind choosing a different experimental setup for the higher dimensional case than the case with $d=3,p=3$, i.e., for why did you set different dimensionality values for single and multiple trajectories? Similar to above, can you please clarify whether (i) you sample a single set of ground-truth parameters or (ii) multiple parameter configurations with different seeds?
> >
> > **2) Real-world examples and the latent DAG assumption.** I appreciate the effort for the examples, but I still do not see a real-world example where the main assumption, DAG structure of the unobserved variables, is satisfied. The population example has no citations. To me, a constant change in environmental factors does not sound realistic. Most likely, the present value of environmental factors would affect the change in the environmental factors. Oscillator example is a linear ODE example from [24] with no unobserved variables. I assume the unobserved variables are added by the authors. Similarly, to me, a constant change in wind (external force) does not sound realistic.

---

> > > ### Author Response · Authors · 2024-08-10
> > > **Response to comment from Reviewer 1wF2**
> > >
> > > Dear Reviewer 1wF2,
> > >
> > > Thank you for your prompt response. We have addressed your concerns as follows.
> > >
> > > **1) Contrived simulation setup.**
> > > - True underlying parameters:
> > >     - The ground-truth parameters are **a single configuration** of $\boldsymbol{\eta}$ as shown in lines 278-279 and the equations below. The identifiable case refers to ground-truth parameters that satisfy our proposed identifiability conditions, while the unidentifiable case involves ground-truth parameters violate these conditions. This configuration of ground-truth parameters is generated randomly rather than being manually designed. In other words, our theoretical results are applicable to any system parameter configurations that meet the proposed identifiability conditions.
> > >     - We conduct $N=100$ replications of experiments for each ground-truth configuration (the identifiable one and the unidentifiable one, respectively) by setting **different initial parameter values through different seeds**. Our simulation aims to verify that, in the identifiable case, parameter estimates from all 100 experiments are consistently close to the ground-truth parameters, as evidenced by the reported results showing low MSE and variance. In the unidentifiable case, some experiments may fit the ground-truth parameters, while others may fit different configurations of system parameters that produce the same observations, leading to higher MSE and variance.
> > > - Our simulation goal is to validate our proposed identifiability conditions. Given the current randomness in initial parameter values, the observed MSE and variance differences between the identifiable and unidentifiable cases strongly support our theoretical results. As previously mentioned, the Nonlinear Least Squares (NLS) loss function used in our simulation is non-convex. Initializing parameter values too far from the true parameters can increase estimation errors due to local minima. These errors stem from the non-global minimizer of the parameter estimation method, not from our theoretical results, which we would like to avoid. Our paper focuses on deriving identifiability condition theories rather than developing parameter estimation methods. The current simulation settings adequately support our theoretical findings.
> > >
> > > - As we have mentioned in our rebuttal to this question,  increasing system dimensions escalates system complexity. For instance, the single trajectory case with $d=3, p=3$ involves $d+p+d^2+d*p+p^2=33$  parameters, whereas the $d=5,p=5$ case involves $85$ parameters. Larger sample sizes and longer estimation times are required to achieve satisfactory parameter estimates in higher dimensions. Due to time constraints, we kept the parameter size moderate, choosing $d=5, p=5$. In the multiple trajectory case,  observations from $p$ trajectories make parameter estimation easier, allowing us to attempt higher dimension such as $d=10, p=5$. As for the configuration, same as the 3-dimensional case, we used a single ground-truth parameter configuration and, due to time limit, conducted $N=10$ replications of experiments for both identifiable and unidentifiable cases.
> > >
> > > **2) Real-world examples and the latent DAG assumption.**
> > >
> > > Thank you for your comment. We have added a citation for the population example for your reference [1]. As you mentioned, these examples are originally fully observable cases, with unobserved variables added by us. Since these systems involve inaccessible variables, the ground-truth model structures are unknow. In such circumstances, researchers or practitioners typically design suitable model structures based on prior experience or relevant physical laws. This is how we incorporated latent variables in these examples.
> > > - For the population example, we consider an environmental factor that changes at a constant rate, such as pollution levels from an industrial plant continuously releasing a fixed amount of pollutants or a wastewater treatment plant releasing a specific amount of treated wastewater into a river hourly.
> > > - For the oscillator example, wind force can be modelled by a constant rate in regions with highly predictable wind patterns, such as during the onset or retreat of monsoon seasons, or in experimental settings where wind force is adjusted at a constant rate. This simple example illustrates that external forces can be modelled with constant derivatives. Additionally, constant forces, or forces with polynomial functions of time $t$, all fit our ODE system structure. For instance, a uniform magnetic field interacting with the system would exert a constant force. These are simple illustrations, and we believe there are many other latent factors that fit well within our ODE structure.
> > >
> > > We hope this addresses your concerns. If you have further questions, please do not hesitate to ask us.
> > >
> > > Sincerely,
> > > The authors
> > >
> > > [1] Mira-Cristiana Anisiu. Lotka, volterra and their model. Did´actica mathematica, 32(01), 2014.

---

> > > > ### Comment · Reviewer_1wF2 · 2024-08-12
> > > >
> > > > Dear authors,
> > > >
> > > > Thank you for the detailed responses. I appreciate the effort to address my concerns, however, as it is, the simulation setup with only a single ground-truth parameter configuration still seems limited. I've decided to keep my score.

---

> ### Author Response · Authors · 2024-08-12
> **Response to comment from Reviewer 1wF2**
>
> Dear Reviewer 1wF2,
>
> Thank you for your response. We appreciate your earnest and responsible approach to reviewing our paper.
>
> However, we believe there may have been some misunderstanding regarding the setup of our simulations. Our simulation methodology follows a standard and widely accepted approach for verifying proposed theories like ours. Specifically, we set a single ground-truth parameter configuration. To ensure reliable and robust parameter estimation results, we then run multiple replications (100 in our simulation) of experiments with different initial parameter values and report the mean and variance of the metric of interest (MSE in our simulation).
>
> We encourage you to take a few minutes to review the simulation settings in the recently accepted JMLR paper [41] and NeurIPS paper [40], which also study the identifiability analysis of linear ODEs and linear SDEs. These papers employ the same simulation setup as ours.
>
> **To address your concerns, we have conducted additional simulations incorporating various ground-truth parameter configurations by utilizing different seeds.** The simulation results strongly affirm the validity of our proposed identifiability conditions. For further details, please refer to the following comment.
>
> Thank you again for your consideration. If you have further questions or need additional clarification, please do not hesitate to contact us.
>
> Sincerely,
>
> The Authors

---

> ### Author Response · Authors · 2024-08-12
> **Response to comment from Reviewer 1wF2**
>
> Dear Reviewer 1wF2,
>
> We would like to reiterate that our theoretical results are applicable to any system parameter configurations that meet the proposed identifiability conditions.
>
> **To further address your concerns, we have added two additional simulations inspired by your comments. Specifically, we set different ground-truth parameter configurations by using different seeds.** Due to time constraints, we have currently applied 10 different configurations to the single and multiple trajectory experiment with $d=3, p=3$. We will update these results to include 100 different configurations in our final manuscript, and we will also include higher-dimensional cases.
>
> The simulation results are provided in the following tables, and they offer strong empirical evidence supporting the validity of our proposed identifiability conditions. Through these additional simulations, we have increased the diversity of our ground-truth examples, providing more convincing empirical support that our proposed theoretical results are suitable for any system parameter configurations that meet the proposed identifiability conditions. We believe that our simulation has been greatly improved by adding this set of simulations.
>
> Table1: MSEs of the $\boldsymbol{\eta}$ -(un)identifiable cases of the ODE (3) with $d=3, p=3$ and **different parameter configurations**
>
> |                  | Identifiable |                     |                      |                        | Unidentifiable |                      |                     |                        |
> | ---------------- | ------------ | ------------------- | -------------------- | ---------------------- | -------------- | -------------------- | ------------------- | ---------------------- |
> | $\boldsymbol{n}$ | $A$          | $B\boldsymbol{z}_0$ | $BG\boldsymbol{z}_0$ | $BG^2\boldsymbol{z}_0$ | $A$            | $BG\boldsymbol{z}_0$ | $B\boldsymbol{z}_0$ | $BG^2\boldsymbol{z}_0$ |
> | 20               | 0.0005       | 2.42E-05            | 0.0038               | 0.0028                 | 0.1309         | 0.2064               | 1.4629              | 0.4528                 |
> |                  | (1.71E-06)   | (3.14E-09)          | (7.91E-05)           | (1.81E-05)             | (0.0276)       | (0.1459)             | (7.6680)            | (1.0949)               |
> | 100              | 0.0001       | 1.36E-05            | 0.0019               | 0.0008                 | 0.0868         | 0.1740               | 0.8929              | 0.1867                 |
> |                  | (3.63E-08)   | (1.14E-09)          | (2.83E-05)           | (5.55E-06)             | (0.0102)       | (0.1019)             | (3.1318)            | (0.1362)               |
> | 500              | 0.0001       | 1.13E-05            | 0.0016               | 0.0005                 | 0.1095         | 0.1788               | 0.7797              | 0.1726                 |
> |                  | (1.64E-08)   | (1.02E-09)          | (1.70E-05)           | (8.33E-07)             | (0.0203)       | (0.1091)             | (3.5265)            | (0.1165)               |
>
>
>
> Table2: MSEs of the $\\{\boldsymbol{\eta}_i\\}_1^p$ -(un)identifiable cases of the ODE (3) with $d=3, p=3$ and **different parameter configurations**
>
> |                  | Identifiable |            |            | Unidentifiable |          |          |
> | ---------------- | ------------ | ---------- | ---------- | -------------- | -------- | -------- |
> | $\boldsymbol{n}$ | $A$          | $B$        | $G$        | $A$            | $B$      | $G$      |
> | 3                | 0.0671       | 0.0972     | 0.1044     | 0.1575         | 0.1019   | 0.1247   |
> |                  | (0.0046)     | (0.0149)   | (0.0092)   | (0.0744)       | (0.0202) | (0.0213) |
> | 10               | 2.53E-07     | 4.98E-09   | 2.06E-08   | 1.4720         | 0.2255   | 0.3048   |
> |                  | (5.76E-13)   | (2.24E-16) | (3.81E-15) | (6.2425)       | (0.3125) | (0.8364) |
> | 20               | 2.24E-08     | 4.42E-10   | 1.83E-09   | 0.7827         | 0.2099   | 1.93E-20 |
> |                  | (4.53E-15)   | (1.76E-18) | (3.00E-17) | (3.5399)       | (0.3193) | (1.91E-39) |
>
> Thank you again for your valuable comments. We hope this additional set of simulations addresses your concerns. If you have further questions, please do not hesitate to ask us.
>
> Sincerely,
>
> The Authors

---

> > ### Comment · Reviewer_1wF2 · 2024-08-12
> >
> > Dear authors,
> >
> > Thanks for the efforts to address my concerns on the simulation setup. I increase my score 4->5.

---

> > > ### Author Response · Authors · 2024-08-12
> > > **Response to comment from Reviewer 1wF2**
> > >
> > > Dear Reviewer 1wF2,
> > >
> > > Thank you very much for raising our score. We greatly appreciate your recognition of our work and your valuable feedback.
> > >
> > > Sincerely,
> > >
> > > The authors

---

### Author Rebuttal · Authors · 2024-08-07

We would like to express our gratitude to all the reviewers for their thoughtful and constructive feedback on our manuscript. Below, we summarize the modifications made to the manuscript based on your comments:

- To Reviewer 1wF2:
    - We have added two real-world linear ODE examples that align well with our interested ODE (2) and (3).
    - We have provided a clear defination of the causal graph in the context of autonomous (time-invariant) ODEs.
    - We have included two higher-dimensional simulation examples.

- To Reviewer Te42
    - We have included a table summarizing all the notations.
    - We have added a table summarizing all the proposed identifiability conditions.
    - We have included two higher-dimensional simulation examples.

- To Reviewer LnTR
    - We have provided explanations for symbols $\boldsymbol{x}_0'$ and $A'$.
    - We have included intuitive explanations for each of our assumptions as they are introduced.
    - We have added two real-world linear ODE examples.

We deeply appreciate the reviewers' insights, which have been invaluable in refining our work.

---

### Decision · Program_Chairs · 2024-09-25

**Decision:**

Accept (poster)

**Comment:**

This paper provides an analysis of identifiability in linear ODE systems incorporating hidden confounders. The reviewers appreciated the theoretical contributions, and the authors provided extensive new experiments in the rebuttal phase in response to reviewer concerns. Authors: please incorporate the promised additions to the paper and revise the paper according to the reviewer suggestions.